# A confined–unconfined aquifer model for subglacial hydrology and its application to the North East Greenland Ice Stream

Sebastian Beyer[1,2], Thomas Kleiner[2], Vadym Aizinger[2,3], Martin Rückamp[2], and Angelika Humbert[2,4]

[1]Potsdam Institute for Climate Impact Research, Potsdam, Germany
[2]Alfred Wegener Institute, Helmholtz Centre for Polar and Marine Research, Bremerhaven, Germany
[3]Friedrich–Alexander University Erlangen–Nürnberg, Erlangen, Germany
[4]University of Bremen, Bremen, Germany

*Correspondence to:* Sebastian Beyer (sebastian.beyer@awi.de)

**Abstract.** Subglacial hydrology plays an important role in ice sheet dynamics as it determines the sliding velocity. It also drives freshwater into the ocean, leading to undercutting of calving fronts by plumes. Modeling subglacial water has been a challenge for decades. Only recently, new approaches have been developed such as representing subglacial channels and thin water sheets by separate layers of variable hydraulic conductivity. We extend this concept by modeling a confined and unconfined aquifer system (CUAS) in a single layer of an equivalent porous medium (EPM). The advantage of this formulation is that it prevents unphysical values of pressure at reasonable computational cost. We performed sensitivity tests to investigate the effect of different model parameters. The strongest influence of model parameters was detected in terms governing the opening and closure of the system. Furthermore, we applied the model to the North East Greenland Ice Stream, where an efficient system independent of seasonal input was identified about $500\,\mathrm{km}$ downstream from the ice divide. Using the effective pressure from the hydrology model, the Ice Sheet System Model (ISSM) showed considerable improvements of modeled velocities in the coastal region.

*Copyright statement.* TEXT

## 1 Introduction

Subglacial water has been identified as a key component in glacial processes, it is fundamental in driving large ice flow variations over short time periods. Recent studies show considerable progress in modeling these subglacial networks and coupling them to ice models. Water pressure strongly influences basal sliding and can therefore be considered a fundamental control on ice velocity and ice-sheet dynamics (Lliboutry, 1968; Röthlisberger, 1972; Gimbert et al., 2016).

Generally, two fundamentally different types of drainage are identified: discrete channel / conduit systems and distributed water sheets or thin films. Distributed flow mechanisms are, for example, linked cavities (Lliboutry, 1968), flows through sediment/till (Hubbard et al., 1995), or thin water sheets (Weertman, 1957); those are considered inefficient and slow. Channels (Röthlisberger, 1969; Shreve, 1972; Nye, 1976) are seen as discrete single features or arborescent networks. They usually

develop over the summer season when a lot of melt water is available. It is assumed that these channelized or efficient drainage systems (able to drain large amounts of water in short time spans) are predominant in alpine glaciers and on the margins of Greenland, where substantial amounts of surface melt water are capable of reaching the bed (van den Broeke et al., 2017). In the interior of Greenland and also in most parts of Antarctica, the water supply is limited to basal melt – a circumstance

favoring distributed systems.

Seasonal variations of ice velocity have been observed and attributed to the evolution of the drainage system switching between an efficient and inefficient state in summer and winter (Bartholomew et al., 2010). For this reason, a new generation of subglacial drainage models has been developed that is capable of coupling the two regimes of drainage and reproducing the transition between them (Schoof, 2010; Hewitt et al., 2012; Hewitt, 2013; Werder et al., 2013; Hoffman and Price, 2014). While

these models demonstrate immense progress for modeling spontaneously evolving channel networks, it is still a challenge to apply them on a  continental scale. A comprehensive overview of the various operational and newly emerging glaciological hydrology models is given in Flowers (2015).

Distributed or sheet structures can naturally be well represented using a continuum approach, while channels usually require a secondary framework, where each feature is described explicitly. Water transport in channels is a complex mechanism that

depends on the balance of melt and ice creep (Nye, 1976; Röthlisberger, 1969), channel geometry, and network topology. Additionally, the network evolves over time which further complicates modeling of this process. When simulating channel networks, particular care must be also taken to prevent the emergence of instabilities due to runaway merging of channels (see the discussion in Schoof et al. (2012)). This leads to increased modeling complexity and high computational costs. An exception to this is the work of de Fleurian et al. (2014), where a sediment layer is used to model the inefficient drainage system (IDS)

and an equivalent porous layer (EPL) represents the efficient drainage of the channel network, both represented by Darcy flow through separate porous media layers. The layer representing the channels has its parameters (namely the hydraulic conductivity and the storage) adjusted to exhibit the behavior of an effective system.

We take this idea even further and apply Darcy flow to only a single layer of an equivalent porous medium (EPM) accounting for both drainage mechanisms (efficient and inefficient) by locally adjusting the effective hydraulic transmissivity. This means

that we approximate the channel flow as a fast diffusion process similarly to work in de Fleurian et al. (2014). Evolution equations based on the development of channels and cavities locally adapt the transmissivity, such that high-transmissivity areas represent the efficient system, while the transmissivity is low for inefficient drainage areas. Similar approaches are known to have been applied to modeling of fracture networks in rock Van Siclen (2002). This reduced complexity model does not capture channels individually but represents their effect by changing specific local properties. We prefer to use the

term "equivalent porous medium" instead of "equivalent porous layer" hereafter to avoid confusion with the terminology in de Fleurian et al. (2014) although both names represent the same approach and are widely used in hydrology. Since our model aims to simultaneously represent the main properties of both drainage mechanisms (efficient and inefficient), special care must be taken when choosing the model parameters and relating them to the physical properties of a specific scenario. In particular, the geometrical and physical parameters used in this model are not directly comparable to observed quantities, but instead

describe an idealized representation that gives the best fit to the available data. While this strategy may not help to advance the

precise understanding of channel formation processes, it captures the overall behavior, is computationally efficient, and allows to examine the complex interactions on larger spatial and temporal scales.

In addition, we differentiate between confined and unconfined flow in the aquifer based on the scheme presented in (Ehlig and Halepaska, 1976). We therefore name our new subglacial hydrology model CUAS (Confined–Unconfined Aquifer Scheme). A sketch with the geometric quantities used in CUAS and the model concept is shown in Fig. 1. While the assumption of always saturated – and therefore confined – aquifers may be true for glaciers with large water supply, it does not hold in areas with lower water input. Especially in locations far from the coast, the water supplies are often insufficient to completely fill the aquifer. Ignoring this leads to significant errors in the computed hydraulic potential and unphysical, i.a. negative, water pressure. This problem has been analyzed in detail by Schoof et al. (2012), but here we study the effect in the context of equivalent media models using unconfined flow as a possible solution.

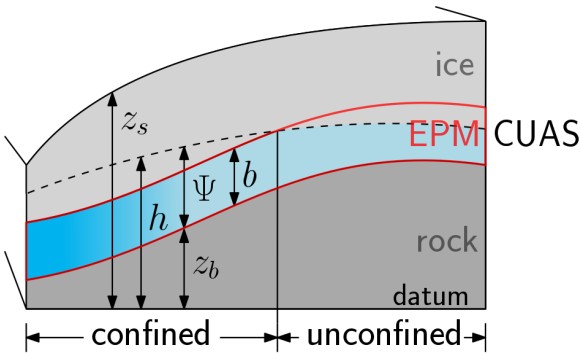

**Figure 1.** Sketch of the EPM model and artificial geometry for experiments. The left side is towards the glacier snout. Red border shows the location of the equivalent porous medium that is modelled. The blue gradient indicates the locally increased transmissivity. When $\Psi < b$ the system becomes unconfined.

Large scale ice flow models often compute the basal velocity using a Weertman-type sliding law, where the inverse of the effective pressure (difference between the ice overburden pressure and the water pressure) determines the velocity at the base. Low effective pressure leads to high basal velocity. Without subglacial hydrology models, the ice models simply take the ice overburden pressure as effective pressure completely neglecting water pressure or absorb all types of pressure into the sliding coefficient for the friction law without explicitly accounting for the contribution of water pressure. This is a major reason why these models struggle to represent fast flowing areas such as ice streams. The effective pressure computed by our model can be easily coupled to an ice sheet model to improve results for fast flowing areas.

Our work is structured as follows. In the next section, we present the one-layer equivalent porous medium model of the subglacial hydrology. In Sect. 3 the model is applied to artificial scenarios, and the sensitivity to model parameters and stability are investigated. In addition, results for seasonal forcing are presented there, and we show how the model evolves over time. Section 4 demonstrates the first application of the proposed methodology to the North East Greenland Ice Stream (NEGIS).

The ice stream penetrates far into the Greenland mainland with its onset close to the ice divide, so sliding apparently plays a major role in its dynamics. A short conclusions and outlook section wraps up the present study.

## 2 Methods

As described above, we chose not to model the efficient and inefficient drainage systems separately, but we use a unified formulation that encompasses both types of water transport in one layer. Our model is based on the assumption that the main characteristics of subglacial hydrology can be captured by an equivalent porous media approach similar to groundwater flow in karstified aquifers (Teutsch and Sauter, 1991). Thus, a Darcy-type groundwater flow equation can be used. This does not mean that we expect the water transport to be through the subglacial sediments, but through an equivalent porous medium, which accounts also for cavities and channels. An appropriate adjustment of its properties can make them capable of exhibiting the same effective transmissivity as e.g. channel systems. The model does not represent water flow through individual channels (which would be represented by Darcy-Weisbach). Instead, we approximate fast flow through the efficient system by Darcy flow with increased transmissivity. We derive the temporal evolution of the controlling parameter —effective transmissivity— from the temporal evolution of the volume occupied by channels (de Fleurian et al., 2016) and cavities (Werder et al., 2013).

The vertically integrated continuity equation in combination with Darcy's law leads to the general *groundwater flow equation* (see e.g. Kolditz et al. (2015)):

$$S\frac{\partial h}{\partial t} = \nabla \cdot (T\nabla h) + Q \tag{1}$$

with $h$ the hydraulic head (water pressure in terms of water surface elevation above an arbitrary datum also known as the piezometric head), $S$ the storage coefficient (change in the volume of stored water per unit change of the hydraulic head over a unit area), $T$ transmissivity, and $Q$ the source term. For a confined aquifer, $T = Kb$, where $K$ is the hydraulic conductivity, and $b$ is the equivalent porous medium thickness. $S = S_s b$ with specific storage $S_s$ given by

$$S_s = \rho_\mathrm{w}\omega g\left(\beta_\mathrm{w} + \frac{\alpha}{\omega}\right) \tag{2}$$

with the acceleration due to gravity $g$, material parameters for the porous medium (porosity $\omega$, compressibility $\alpha$) and water (density $\rho_\mathrm{w}$, compressibility $\beta_\mathrm{w}$).

Water pressure $P_\mathrm{w}$ and effective pressure $N$ are related to hydraulic head as

$$P_\mathrm{w} = \Psi\rho_\mathrm{w}g \tag{3}$$

and

$$N = P_\mathrm{i} - P_\mathrm{w} \tag{4}$$

with $\Psi = h - z_b$ the local height of the head over bedrock $z_b$ and $P_\mathrm{i} = \rho_\mathrm{i}gH$ the cryostatic ice overburden pressure exerted by ice with thickness $H$ and density $\rho_\mathrm{i}$ (see Fig. 1).

## 2.1 Opening and closure

Opening and closure of channels is governed by the melt at the walls due to the dissipation of heat and the pressure difference between the inside and outside of the channel leading to creep deformation. Linked cavities open due to sliding over bedrock bumps (Walder, 1986; Kamb, 1987). Most existing models use separate descriptions for the efficient and the inefficient transport system (e.g. continuum description for sheet-flow and discrete channels) leading to two sets of equations that need to be coupled. Our single layer medium allows us to use a single set of equations that includes melt opening, cavity opening and creep closure, which is quite compelling given that channels and sheets are only the extremes of a much more varied drainage system. In this regard, our model is similar to the one by Schoof (2010), though we use a continuum description, which can cause instabilities (runaway growth) when the melt rate is much larger than the creep closure (Hewitt, 2011). However, the diffusive nature of our model avoids this problem by distributing the growth over a small area, thus preventing infinite growth and leading to a stable configuration.

We adopt the classical channel equations from Nye (1976) and Röthlisberger (1972) as in de Fleurian et al. (2016) and cavity opening (Walder, 1986; Kamb, 1987) as in Werder et al. (2013) to evolve the effective transmissivity. The details on this are shown in Appendix A. Thus

$$
\frac{\partial T}{\partial t} = \underbrace{\frac{g\rho_{\mathrm{w}} K T}{\rho_{\mathrm{i}} L}(\nabla h)^2}_{\text{melt}} - \underbrace{2An^{-n}|N|^{n-1}NT}_{\text{creep}} + \underbrace{\beta|\mathbf{v}_b|K}_{\text{cavities}},
\tag{5}
$$

with $L$ the latent heat, $\beta$ a factor governing opening via sliding over bedrock protrusions, $\mathbf{v}_b$ basal velocity of the ice, $A$ the creep rate factor depending on temperature, and $n$ the creep exponent, which we choose as $n = 3$. The dimensionless parameter $\beta = b_r/l_r$ depends on the height $b_r$ and distance $L_r$ of the bedrock protrusions. The cavity opening formulation does not yet include a limit imposed by the bump height. Depending on the sign of $N$, creep closure as well as creep opening can occur. Negative effective pressure over prolonged time is usually considered unphysical, and the correct solution to this would be to allow the ice to separate from the bed (see e.g. Schoof et al. (2012) for a possible solution). However, in the context of our equivalent layer model, the creep term in Eq. (5) is still applicable because this is how a channel would behave for $N < 0$. In Sect. 3.1, we test the sensitivity of $T$ and $N$ to the magnitudes of $K$, $\beta$, and $A$.

## 2.2 Confined–Unconfined Aquifer Scheme

The water balance equation (Eq. 1) and the pressure equation (Eq. 3) assume that the porous medium is always completely filled with water. As this is not always true, especially for areas with significant bedrock topography combined with low water input, it is possible to obtain unphysical negative water pressures with this method. A possible solution is to relax the assumption of an always filled medium and consider the general form (confined and unconfined). We follow Ehlig and Halepaska (1976) and write the general form for the confined–unconfined problem:

$$
S_e(h)\frac{\partial h}{\partial t} = \nabla \cdot (T_e(h)\nabla h) + Q.
\tag{6}
$$

Now the effective transmissivity $T_e$ and the effective storage coefficient $S_e$ depend on the head and are defined as

$$T_e(h) = \begin{cases} T, & b \leq \Psi \quad \text{confined} \\ K\Psi, & b > \Psi \quad \text{unconfined} \end{cases} \tag{7}$$

and

$$S_e(h) = S_s b + S'(h) \tag{8}$$

with

$$S'(h) = \begin{cases} 0, & b \leq \Psi & \text{confined,} \\ (S_y/d)(b - \Psi), & b - d \leq \Psi < b & \text{transition,} \\ S_y, & 0 \leq \Psi < b - d & \text{unconfined.} \end{cases} \tag{9}$$

This means that as soon as the head sinks below the aquifer height, the system becomes unconfined, and therefore only the saturated section contributes to the transmissivity calculation. This also prevents the head from falling below the bedrock as detailed in Section 3.2. Additionally, the mechanism for water storage changes from elastic relaxation of the aquifer (confined) to dewatering under the forces of gravity (unconfined). The amount of water released from dewatering is described by the specific yield $S_y$. Since this amount is usually orders of magnitudes larger than the release from the confined aquifer ($S_y \gg S_s b$), it is useful to introduce a gradual transition as in Eq. (9) controlled by a user defined transition parameter $d$. At each time step, the model solves the equation for hydraulic head (Eq. 6) and evolves the transmissivity of the EPM according to Equation 5. Note that the transmissivity is not homogeneous making Eq. (6) nonlinear. This fits with our approach to describe the effective system (channels) by locally increasing the transmissivity. The drawback of this formulation is that the evolution of $T$ does not affect areas where the flow is unconfined (as $T_e = K\Psi$ for unconfined). Also, the melt rate for the opening term (Equation 5) does not account for the possibility of unconfined flow. This is not an issue because unconfined flow occurs only in the locations where the water supply is low, i.e., where no channels are expected to develop. Details on the numerical implementation can be found in Appendix B. The benefit of this approach is discussed in Sect. 3.2.

## 3 Experiments with artificial geometries

Testing the EPM concept and determining parameters is not straightforward because there are no directly comparable physical properties. Moreover, observations and measurements of subglacial processes are in general difficult and sparse. We address this by testing the model with some of the benchmark experiments of the **S**ubglacial **H**ydrology **M**odel **I**nter-comparison **P**roject (De Fleurian et al., 2018). The proposed artificial geometry mimics a land-terminating ice sheet margin measured $100\,\text{km}$ in the $x$-direction and $20\,\text{km}$ in the $y$-direction. The bedrock is flat ($z_b(x,y) = 0\,\text{m}$) with the terminus located at $x = 0\,\text{km}$, while the surface $z_s$ is defined by a square root function $z_s(x,y) = 6\left((x + 5000)^{1/2} - 5000^{1/2}\right) + 1$. Here, we use the SHMIP/B2 setup, which includes 10 moulins with steady supply. Boundary conditions are set to zero influx at the interior boundaries ($y = 0\,\text{km}$, $y = 20\,\text{km}$, $x = 100\,\text{km}$) and zero effective pressure at the terminus. All experiments start with the initial conditions that imply zero effective pressure and are run for 50 years to ensure that a steady state is reached.

## 3.1 Parameter estimation and sensitivity

SHMIP is primarily intended as a qualitative comparison between different subglacial hydrology models, where results from the GlaDS model (Werder et al., 2013) serve as a "common ground". Here, we use it as a basis for an initial tuning and a study of the sensitivity of our model with regard to parameters. SHMIP is presenting an in-depth comparison of all models, which is also the reason why we do not show a comparison to other models in this study.

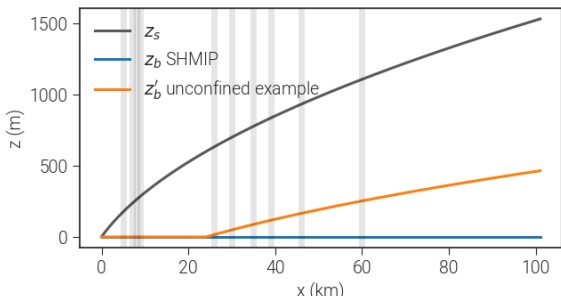

**Figure 2.** Experiments with artificial geometries. Vertical lines denote moulin positions for SHMIP/B2. The orange line shows the modified bedrock used to illustrate the impact of the confined/unconfined scheme as discussed in Sect. 3.2

**Table 1.** Physical constants used in the model. We distinguish between well known (upper half) and estimated / uncertain (lower half) parameters.

| Name | Definition | Value | Units |
|------|-----------|-------|-------|
| $L$ | latent heat of fusion | 334 | $\mathrm{kJ\,kg^{-1}}$ |
| $\rho_{\mathrm{w}}$ | density of water | 1000 | $\mathrm{kg\,m^{-3}}$ |
| $\rho_{\mathrm{i}}$ | density of ice | 910 | $\mathrm{kg\,m^{-3}}$ |
| $n$ | flow law exponent | 3 | - |
| $g$ | gravitational acceleration | 9.81 | $\mathrm{m\,s^{-2}}$ |
| $\beta_{\mathrm{w}}$ | compressibility of water [a] | $5.04 \times 10^{-10}$ | $\mathrm{Pa^{-1}}$ |
| $\alpha$ | compressibility of porous medium[a] | $10^{-8}$ | $\mathrm{Pa^{-1}}$ |
| $\omega$ | porosity[a] | 0.4 | - |
| $S_s$ | specific storage, Eq. (8) | $\approx 1 \times 10^{-3}$ | $\mathrm{m^{-1}}$ |
| $S_y$ | specific yield | 0.4 | |

[a] Values from de Fleurian et al. (2014)

**Table 2.** Model parameters (upper) and variables computed in the model (lower)

| Name | Definition | Units |
|---|---|---|
| $T_{\mathrm{min}}$ | min. transmissivity | $\mathrm{m^2\,s^{-1}}$ |
| $T_{\mathrm{max}}$ | max. transmissivity | $\mathrm{m^2\,s^{-1}}$ |
| $b$ | equivalent porous medium thickness | m |
| $d$ | confined / unconfined transition (Eq. (9)) | m |
| $Q$ | water supply | $\mathrm{m\,s^{-1}}$ |
| $A$ | creep rate factor | $\mathrm{Pa^{-3}\,s^{-1}}$ |
| $K$ | hydraulic transmissivity | $\mathrm{m\,s^{-1}}$ |
| $\mathbf{v}_b$ | basal ice velocity | $\mathrm{m\,s^{-1}}$ |
| $\beta$ | cavity opening parameter | |
| $h$ | hydraulic head | m |
| $S$ | storage | - |
| $S_e$ | effective storage | - |
| $T_e$ | effective transmissivity | $\mathrm{m^2\,s^{-1}}$ |
| $T$ | transmissivity | $\mathrm{m^2\,s^{-1}}$ |
| $a_{\mathrm{melt}}$ | opening by melt | $\mathrm{m^2\,s^{-2}}$ |
| $a_{\mathrm{cavity}}$ | opening by sliding over bedrock | $\mathrm{m^2\,s^{-2}}$ |
| $a_{\mathrm{creep}}$ | opening/closure by creep | $\mathrm{m^2\,s^{-2}}$ |
| $P_{\mathrm{w}}$ | water pressure | Pa |
| $P_{\mathrm{i}}$ | ice pressure | Pa |
| $N$ | effective pressure | Pa |

In Table 1, we show the physical constants used in all setups and runs. The values in the lower half are properties of the porous medium and are only estimated. Since they are utilized in the context of the EPM concept this is not an issue. Table 2 contains the model parameters in the upper part and the variables computed by the model in the lower part.

We divide the sensitivity analysis into a general block investigating the sensitivity to the amount of water input into moulins, the layer thickness $b$, the confined / unconfined transition parameter $d$, grid resolution $dx$ (Fig. 3) and a block that examines the parameters directly affecting channel evolution such as creep rate factor $A$, conductivity $K$, and the bounds for the allowed transmissivity $T_{\mathrm{min}}$ and $T_{\mathrm{max}}$ (Fig. 4). In Table 3, we present values that lead to the best agreement with the SHMIP benchmark experiments and thus are used in the following as the baseline for our sensitivity tests.

In Figs. 3a and b, the model's reaction to different amounts of water input through the moulins is shown. With deactivated transmissivity evolution ($T = \mathrm{const.}$, dashed lines), larger water inputs lead to higher water pressure, hence lower effective pressure $N$. In this case, a moulin input of $18\,\mathrm{m^3\,s^{-1}}$ leads to negative values of $N$. With activated evolution of $T$, the transmissivity adapts to the water input: as more water enters the system through moulins, the transmissivity rises. Vertical gray bars

**Table 3.** Selected baseline parameters for all experiments unless otherwise noted. These parameters best match the SHMIP targets.

| Name | Value | Units |
|------|-------|-------|
| $T_{\min}$ | $1 \times 10^{-7}$ | $\mathrm{m^2\,s^{-1}}$ |
| $T_{\max}$ | $100$ | $\mathrm{m^2\,s^{-1}}$ |
| $b$ | $0.1$ | $\mathrm{m}$ |
| $d$ | $0$ | $\mathrm{m}$ |
| $dx$ | $1000$ | $\mathrm{m}$ |
| $A$ | $5 \times 10^{-25}$ | $\mathrm{Pa^{-3}\,s^{-1}}$ |
| $K$ | $10$ | $\mathrm{m\,s^{-1}}$ |
| $\beta$ | $5 \times 10^{-4}$ | |
| $Q_{\text{per moulin}}$ | $9$ | $\mathrm{m^3\,s^{-1}}$ |

show the location of moulins along the x-axis, and the most significant increase in $T$ occurs directly downstream of a moulin. This happens because the water is transported in this direction leading to increased melt. At the glacier snout ($x = 0$), the ice thickness is at its lowest so almost no creep closure takes place; hence, the transmissivity grows large for all tested parameter combinations. Significant development of effective drainage is visible for inputs above $0.07\,\mathrm{m^3\,s^{-1}}$ (yellow line). The resulting

effective pressure decreases with rising water input as the system becomes more efficient at removing water. Up to ca. $35\,\mathrm{km}$ distance from the snout this results in similar values of $N$ for all forcings above $0.28\,\mathrm{m^3\,s^{-1}}$. The system adapts so that it can remove all of the additional water efficiently. In Figs. 3i and j, the two-dimensional distributions of $N$ and $T$ are shown for the baseline parameters. In the following, we denote regions of high transmissivity as channels, even though our model does not directly simulates them. Those channels form downstream from moulins and continue straight towards the ocean. The effective

pressure drops around water inputs and along the channels. We observe no sensitivity of our result to the layer thickness $b$ (Figs. 3c and d). Because we use transmissivity, $b$ does not influence the flow of water directly, but is important to decide when the system becomes unconfined, as well as determining the Storage (see Eq. 8). However, in this experiment the system has sufficient water input so that all cells are confined in the steady state and also the storage has not influence on the long time solution (The storage determines how fast a pressure change travels through the system, but is irrelevant for the steady state).

The large availability of water also explains why the confined–unconfined transition parameter $d$ does not show noticeable effects on the results (Figs. 3e and f) – the system is always confined. Grid resolution $dx$ has low influence on the pressure distribution and a minor effect on the transmissivity downstream (Figs. 3g and h). However, coarse resolutions are unable to resolve the steps that appear at the moulins.

In Figs. 4a and b, we show the results for different values of $T_{\min}$. $T_{\min}$ acts as a numerical limit to avoid infinite growths

for ill-posed conditions and do generally not show influence on the results. If $T_{\min}$ is chosen very large ($0.1\,\mathrm{m^2\,s^{-1}}$ or larger), this dominates the balance between opening and closure and leads to high water flux, increasing the effective pressure. $T_{\max}$ (Fig. 4b and c) has no visible impact on the resulting pressure distribution. The creep rate factor $A$ determines the "softness"

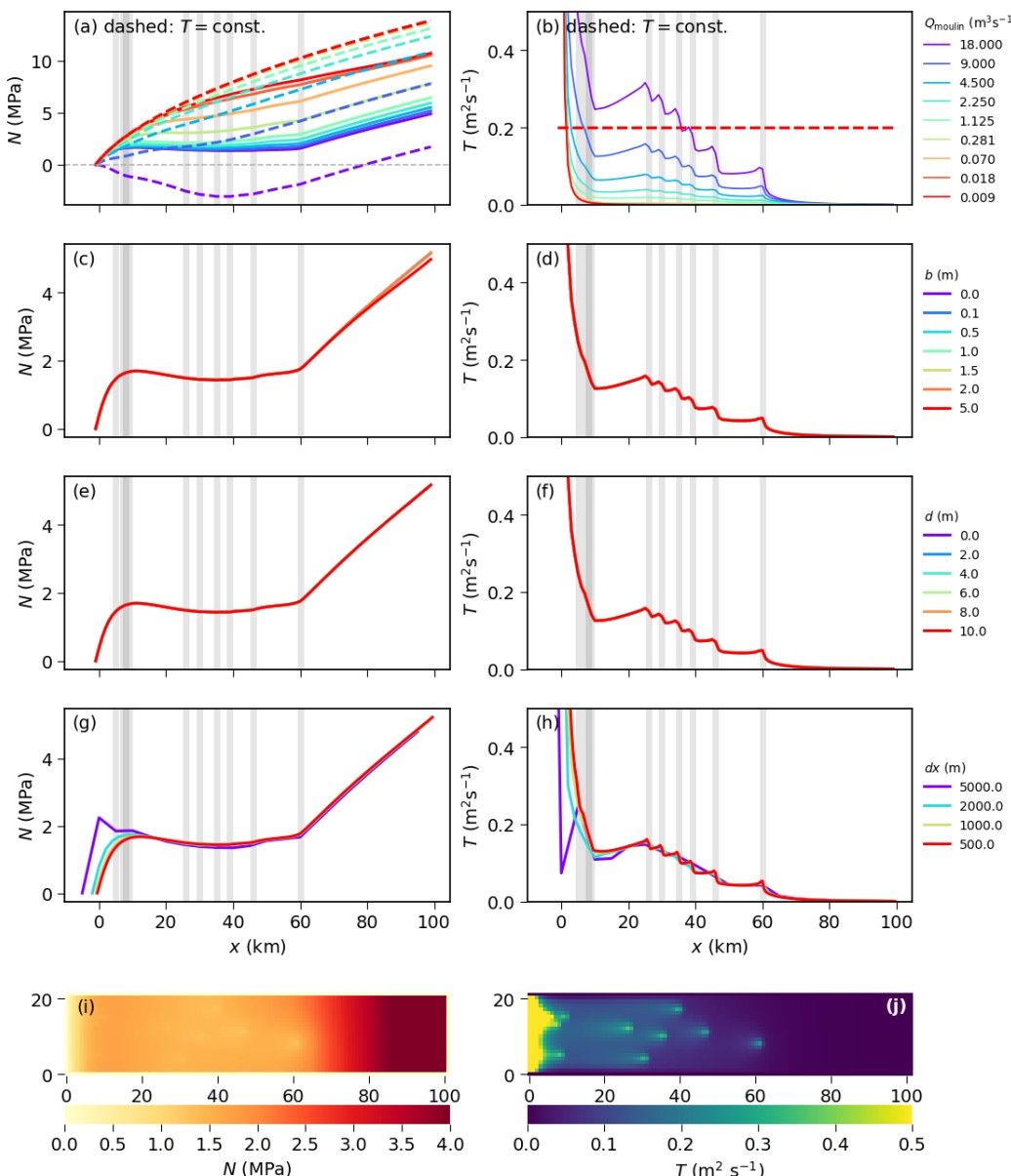

**Figure 3.** Results from the general sensitivity experiments showing the dependence of $N$ (left) and $T$ (right) on: (a)–(b) Water supply from moulins $Q_{\mathrm{moulin}}$ (results for deactivated transmissivity evolution are shown using dashed lines), (c)–(d) aquifer layer thickness $b$, (e)–(f) confined/unconfined transition parameter $d$, (g)–(h) grid resolution $dx$. Shown values are averaged along the y-axis to represent cross-sections at flow lines. Transmissivity plots are cut off at $0.5\,\mathrm{m^2 s^{-1}}$ to improve visibility of the relevant range. (i) and (j) show the two-dimensional distributions (map view) of the results using the best-fit baseline parameters.

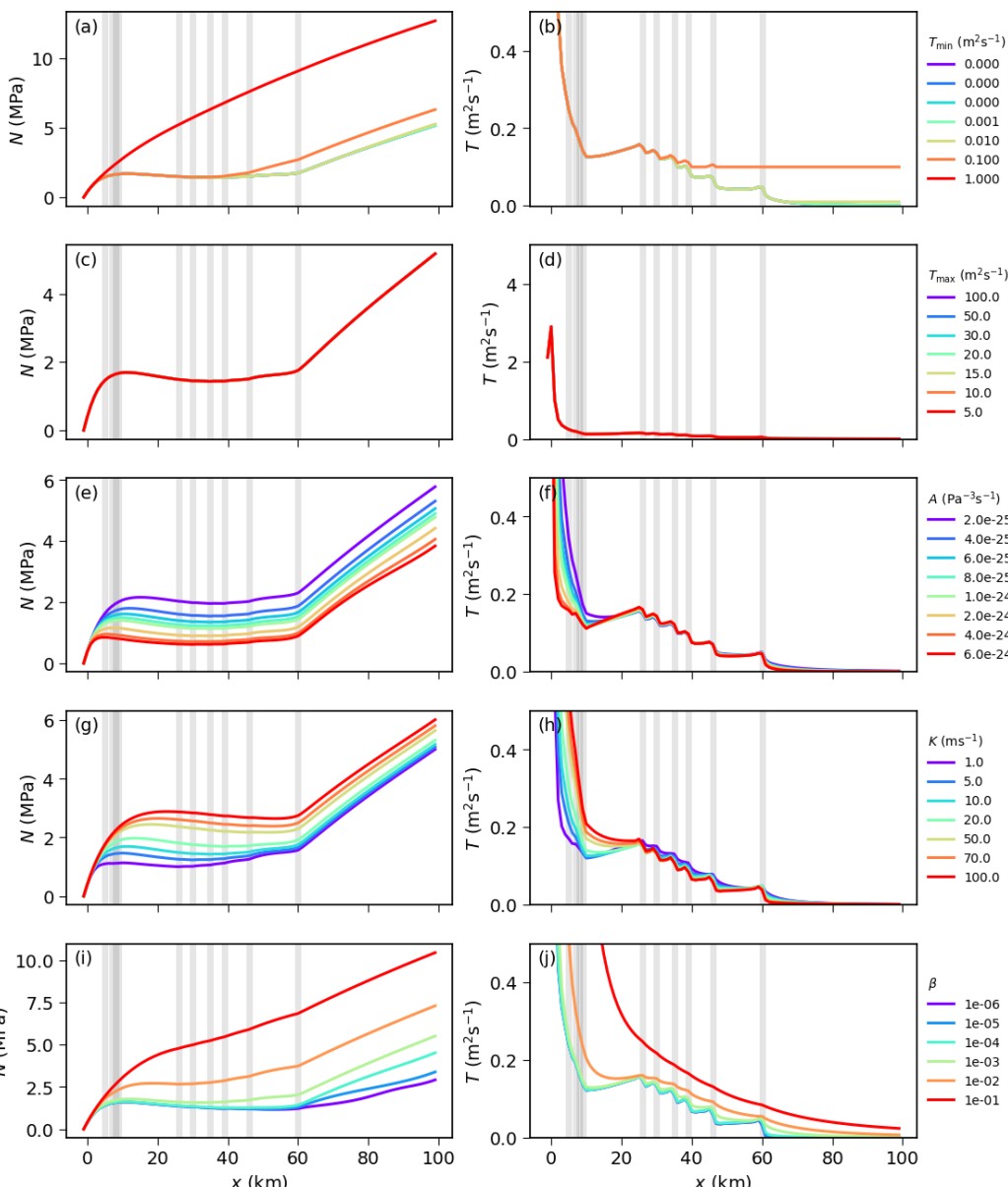

**Figure 4.** Results from parameters directly related to opening and closure: Limits on the transmissivity $T_{\min}$ (panels a and b) and $T_{\max}$ (panels c and d), creep rate factor $A$ (panels e and f), conductivity $K$ (panels g and h) and cavtity opening parameter$\beta$ (panels i and j). Shown values are averaged along the y-axis to represent cross-sections at flow lines. Transmissivity plots are cut off at $0.5\,\mathrm{m^2s^{-1}}$ to improve visibility of the relevant range.

of the ice and therefore affects the creep term in Eq. (5). Larger values of $A$ imply warmer ice; hence, more creep closure (see Figs. 4e and f). Note, that this also affects creep opening for $N < 0$. The conductivity $K$ describes the flux of water through

the system and therefore determines the melt term (see Eq. 5). Larger values of $K$ lead to higher transmissivity and more water transport resulting in lower $P_w$ and higher $N$.

In order to explore the solution dependence on cavity evolution, we assume the basal ice velocity $\mathbf{v}_b = 1 \times 10^{-6} \, \mathrm{m \, s^{-1}}$ (as in SHMIP) and vary $\beta$. $\beta$ parametrizes the bedrock geometry and incorporates height and distance of protrusions. As expected, larger values of $\beta$ lead to more opening and, therefore, a higher effective pressure. With values as high as $1 \times 10^{-1}$, the cavity opening completely dominates the transmissivity evolution, and the effect of moulins is not visible anymore.

## 3.2 The benefit from treating unconfined aquifer

As described above, the confined–unconfined aquifer approach is advantageous for obtaining physically meaningful pressure distributions. In the example illustrated in Fig. 5, we use a slightly modified geometry, where the bedrock rises towards the upstream boundary forming a slab $z'_b(x,y) = \max \left( 3((x+5000)^{1/2} - 5000^{1/2}) - 300, 0 \right)$. The supply is constant in time and space, and we choose a low value of $7.93 \times 10^{-11} \, \mathrm{m \, s^{-1}}$ ($\approx 2.5$ mm/a) to compare our improved scheme to the simple confined only case. Fig. 5 shows a comparison of the steady state solutions: For the confined-only case, the hydraulic head drops below the bedrock at the upstream region. This results in negative water pressure for these regions. Addressing this by simply limiting the water pressure to zero would result in inconsistencies between the pressure field and the water supply. Our new scheme limits the transmissivity when the head approaches the bedrock and by this means ensures $p_w \geq 0$ in a physically consistent way. Additionally, the confined-only solution completely depends on boundary conditions and supply terms, basal topography has no influence in this case (apart from governing $dK/dt$). The possibility of the aquifer to become unconfined captures the expected behaviour much better: At high water levels, water pressure distribution dominates water transport, while at low levels the bed topography becomes relevant.

## 3.3 Seasonal channel evolution and properties

In order to understand our model's ability to simulate the seasonal evolution of subglacial systems, we selected the setup SHMIP/D and ran it with different values of key model parameters. This experiment does not include any moulins but prescribes a non-uniform spatial distribution of supply instead that also varies seasonally. A simple degree day model with varying temperature parameter $d\Theta$ provides water input rising from the downstream end (lowest elevated) of the glacier towards the higher elevated areas over summer:

$$\Theta(t) = -16\cos\left(2\pi/\mathrm{yr} \, t\right) - 5 + d\Theta \tag{10}$$

$$Q_{\mathrm{dist}}(z_s, t) = \max\left(0, (z_s \mathrm{LR} + \Theta(t)) \mathrm{DDF}\right) + Q_{\mathrm{basal}}. \tag{11}$$

Here, $\mathrm{yr} = 31536000 \, \mathrm{s}$ denotes the number of seconds per year, $\mathrm{LR} = -0.0075 \, \mathrm{K \, m^{-1}}$ the lapse rate, $\mathrm{DDF} = 0.01/86400 \, \mathrm{m \, K^{-1} \, s^{-1}}$ is the degree day factor, and $Q_{\mathrm{basal}} = 7.93 \times 10^{-11} \, \mathrm{m \, s^{-1}}$ represents additional basal melt. The resulting seasonal evolution of the supply is shown in Fig.6a. The model is run for 10 years so that a periodic evolution of the hydraulic forcing is generated. Here, we present the result for one parameter set only since the model is not very sensitive in this setup.

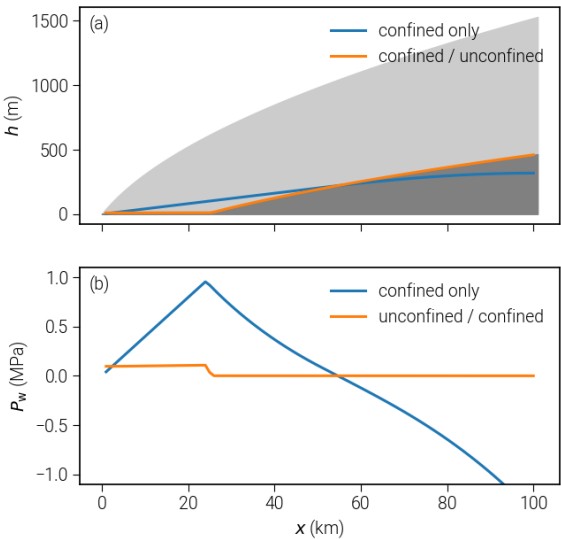

**Figure 5.** Advantages of using the confined/unconfined aquifer scheme (CUAS): Values of head and water pressure for geometries with non-flat bedrock. (a) Computed head for the confined and combined scheme with ice geometry in the background. In the confined only case, the head goes below bedrock. (b) Resulting water pressure, only for the combined scheme the pressure is always non-negative.

We chose three different locations to present $N$ and $T$ during the season: downstream of the glacier close to the snout, in the center, and at a far upstream location (Figs. 6b–d; the locations are marked in panel g). The time series are spatially averaged over these locations with solid lines representing the effective pressure and dashed lines the transmissivity. Water input increases during the summer months, while the corresponding effective pressure drops. With a time lag the transmissivity

rises in response. Supply develops from downstream towards the upstream end of the glacier over the season so the decline in $N$ at the downstream location (Fig. 6b) is instantaneous when the supply rises, while at the further inland locations (Figs. 6c and d), $N$ reacts later during the year. At the middle location, the drop in $N$ is only visible for temperature parameters of -2 and higher. The rise in transmissivity occurs for the three highest temperatures. Finally, at the upstream position, only for $d\Theta = 4$ and $d\Theta = 2$ the effective pressure drops below zero, while for $d\Theta = 0$ the drop is smaller in magnitude and more prolonged.

The transmissivity rise is only significant for $d\Theta = 4$ at this location. While the onset and minima of the decline in $N$ strongly depend on the amount and timing of the water input for all values of $d\Theta$, the maximum of $T$ and also the time when $N$ returns to winter conditions is similar. For the downstream position, the maximum transmissivity is reached for day 210 (not visible in the figure), and $N$ reaches its background value approximately 25 days later. At the center and upstream positions, this behavior is less pronounced but generally similar.

The observed behavior is expected and indicates that our model is able to represent the seasonal evolution of the subglacial water system. Increasing water supply over the year leads to rising water pressure and dropping effective pressure. When the transmissivity rises in response, the effective pressure goes up again despite the supply not yet falling again because the more efficient system is able to transport the water away. For the cases, where no visible change in $T$ occurs such as $d\Theta = -6$

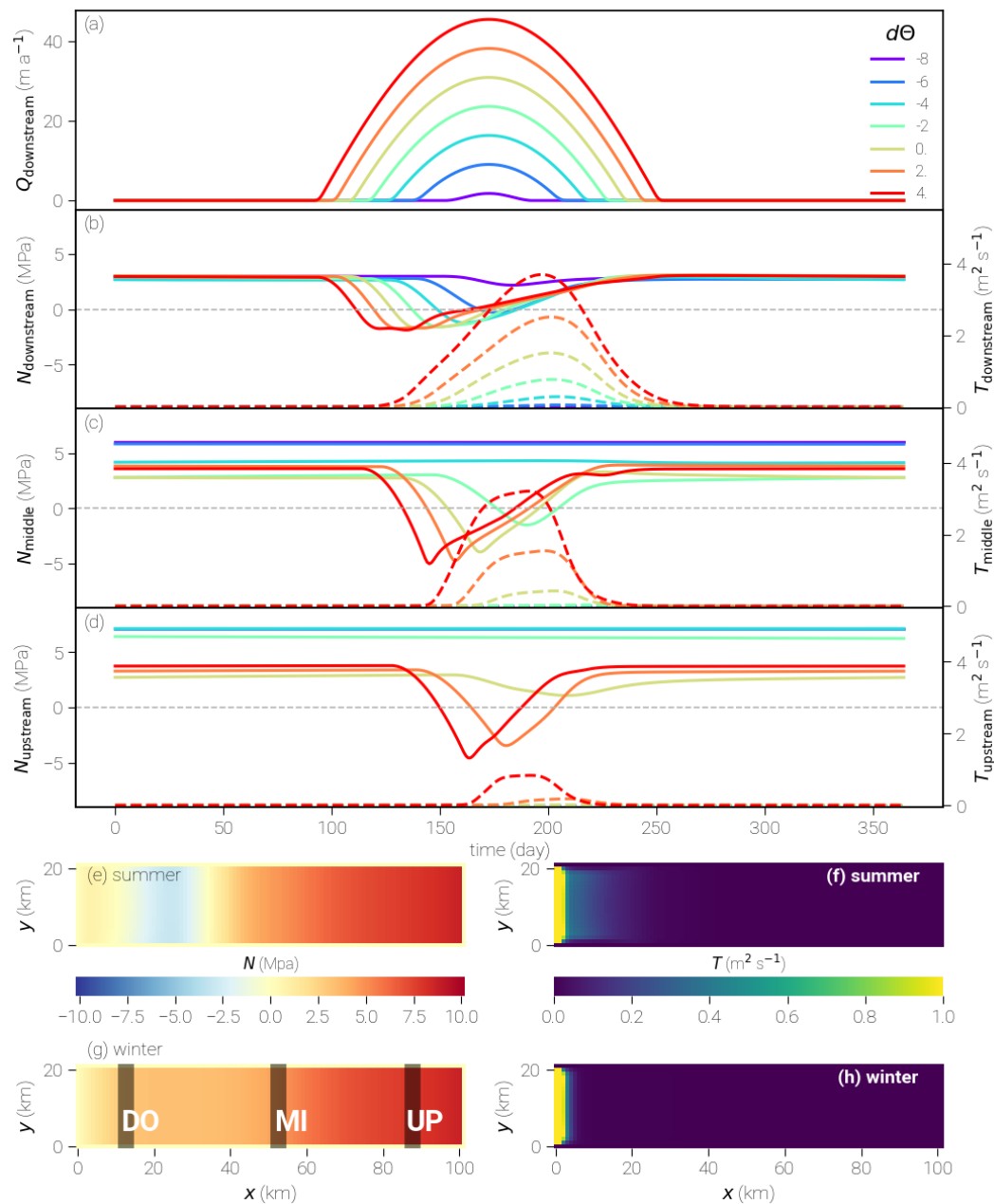

**Figure 6.** Results for one season of the SHMIP/D experiment. In panels (b)–(d), the left axis (effective pressure) corresponds to the solid lines, while the right axis (transmissivity) specifies the values for the dashed lines. The values at the given positions (upstream, middle, downstream) are averaged over the corresponding areas indicated in panel (g). Panels (e)–(h) show two-dimensional distribution maps of $d\Theta = -4$ run.

(blue line in Fig. 6b), the effective pressure follows the supply at the terminus with a small delay, while at the center position ($d\Theta = -2$, cyan line, Fig. 6c), the minimum is offset by the time needed for the supply to reach that location. The maximum in transmissivity $T$ is reached later because, once the system becomes efficient, increased water transport stimulates melting that opens the system even more. This self-reinforcing process is only stopped when enough water is removed and the reduced

water flux reduces the melt again. We assume that this leads to similar locations of the transmissivity maxima for different $d\Theta$ and the resulting similar reemerging of winter conditions in $N$.

In this experiment, $N$ becomes negative during the seasonal evolution, which is not physically meaningful. We attribute such behavior to a lack of adjustment of water supply to the state of the system. In reality, the supply from runoff or supraglacial drainage would cease as soon as the pressure in the subglacial water system becomes too high; here we simply continue to

pump water into the subglacial system without any feedback. This then leads to negative values of $N$. It is also consistent with the finding that $N$ becomes negative earlier in the season in cases of higher supply. We will address this deficiency in future work.

## 4   Subglacial hydrology of NEGIS, Greenland

The role of subglacial hydrology in the genesis of ice streams is yet not well understood. NEGIS is a very distinct feature of the

ice sheet dynamics in Greenland; thus, the question about the role of subglacial water in the genesis of NEGIS is critical. The characteristic increase in horizontal velocities becomes apparent about $100\,\mathrm{km}$ downstream from the ice divide (Vallelonga et al., 2014). Further downstream, the ice stream splits into three different branches: the 79° North Glacier (79NG), Zacharias Isbrae (ZI), and Storstrømmen. Thus far, large scale ice models have only been able to capture the distinct flow pattern of NEGIS when using data assimilation techniques such as inverting for the basal friction coefficient (see e.g. horizontal velocity

fields in Goelzer et al., 2018). It is assumed that most of the surface velocity can be attributed to basal sliding amplified by basal water instead of ice deformation (Joughin et al., 2001). This means that the addition of a subglacial hydrology might have the potential to improve the results considerably. While many glaciers in Greenland have regularly draining supraglacial lakes and run-off driving a seasonality of the flow velocities, little is known about the effect at NEGIS (Hill et al., 2017). Because of this lack of data, to avoid an increased complexity, and to focus on the question if basal melt alone can account for the development

of an efficient system, we do not include any seasonal forcing into our experiment. Our setup includes the major parts of this system. The pressure-adjusted basal temperature $\Theta_{\mathrm{pmp}}$ obtained from PISM (Aschwanden et al., 2016) is utilized to define the modeling region. We assume that for freezing conditions at the base ($T_{\mathrm{pmp}} < 0.1\,\mathrm{K}$) basal water transport is inhibited and take this as the outline of our model domain. Fig. 7 shows the selected area and PISM basal melt rates used as forcing. For the ice geometry, we use the bed model of Morlighem et al. (2014) interpolated on a $1.2\,\mathrm{km}$ grid. Boundary conditions at lateral

margins are set to no flux, whereas the termini at grounding lines are defined as Dirichlet boundaries with a prescribed head that implies an effective pressure of zero. This means that the water pressure at the terminus is equal to the hydrostatic water pressure of the ocean assuming floating condition for the ice at the grounding line. Parameters used for this experiment are

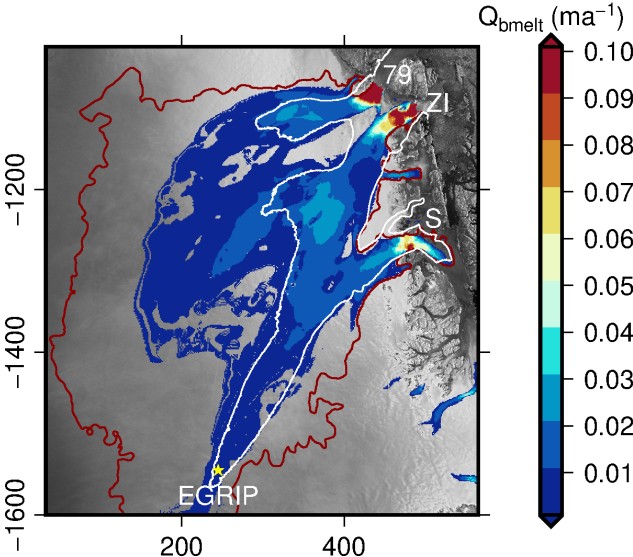

**Figure 7.** Boundary conditions and forcing for NEGIS experiment. Shown is the basal melt rate from PISM and contour line for $\Theta_{\mathrm{pmp}} = -0.1\,\mathrm{K}$ (red) used as model boundary. The white line indicates the $50\,\mathrm{m\,a^{-1}}$ velocity contour.

the same as in our sensitivity study (Table 3). The simulation is run for $50\,\mathrm{a}$ to reach steady state. Despite a high resolution ($444 \times 481$), computing time for this setup is still reasonable (3.5 hours on a single core of Intel Xeon Broadwell E5-2697).

The resulting distributions of effective pressure and transmissivity are shown in Figs. 8a and b, respectively. As expected, effective pressure is highest at the ice divide and decreases towards the glacier termini. Transmissivity is low for the majority

of the study area with the exception of the vicinity of grounding lines and two distinct areas that touch in between 79NG and ZI. The northern area (marked I in Fig. 8b) is located at the northern branch of 79NG and has no direct connection to the snout. The second area (marked II in Fig. 8b) emerges in the transition zone between the southern branch of 79NG and ZI and covers an area approximately twice as large as area I with higher values of $T$. It reaches down to the snout of ZI.

Comparing the effective pressure distribution to the observed velocity (Rignot and Mouginot, 2012) – we chose the $50\,\mathrm{m\,a^{-1}}$

contour line as indicator of fast flow – we observe a high degree of overlap between the fast flowing regions and those with low effective pressure (below $1\,\mathrm{MPa}$) over most of the downstream domain of our study area. Storstrømmen shows higher effective pressure downstream than 79NG and ZI, which is in accordance with lower observed horizontal velocities for that glacier (Joughin et al., 2010). At the onset of the NEGIS, the effective pressure is high, and no relationship to the flow velocity can be observed.

To further examine the possible influence of our hydrology model to basal sliding, we investigate the impact on the sliding law. We chose to compare our computed $N_{\mathrm{CUAS}}$ to the reduced ice overburden pressure defined in Huybrechts (1990) as $N_{\mathrm{HUY}} = P_{\mathrm{i}} + \rho_{\mathrm{sw}}g(z_b - z_{\mathrm{sl}})$ for $z_b < z_{\mathrm{sl}}$ and $N_{\mathrm{HUY}} = P_{\mathrm{i}}$ otherwise. The quotient of $N_{\mathrm{HUY}}$ to $N_{\mathrm{CUAS}}$ is shown in Fig. 8c to demonstrate where the application of our hydrology model would increase basal velocities.

In order to demonstrate the effect of the modeled subglacial hydrology system on the NEGIS ice flow, we setup a simple, one-way coupling to an ice flow model. Here, we use the Ice Sheet System Model (ISSM, Larour et al., 2012), an open source finite element flow model appropriate for continental scale and outlet glacier applications (e.g. Bondzio et al., 2017; Morlighem et al., 2016). The modeling domain covers the grounded part of the whole NEGIS drainage basin. The ice flow is

modeled by the higher order approximation (HO, Blatter, 1995; Pattyn, 2003) in a 3D model, which accounts for transversal and longitudinal stress gradients. In the HO-model we do not perform a thermo-mechanical coupling, but prescribe a depth-averaged hardness factor in Glens flow law instead. Model calculations are performed on an unstructured finite element grid with a resolution of 1 km in fast flow regions and of 20 km in the interior. Basal drag $\boldsymbol{\tau}_b$ is determined by a Budd sliding law:

$$\boldsymbol{\tau}_b = -k^2 N \mathbf{v}_b, \tag{12}$$

where $k^2$ is a positive constant. We run two different scenarios, where (1) the effective pressure is parametrized as the reduced ice overburden pressure, $N = N_{\mathrm{HUY}}$, and (2) the effective pressure distribution is taken from the hydrological model at steady state, $N = N_{\mathrm{CUAS}}$. The value of $k^2$ is tuned in order to have ice velocities of approximately $1500\,\mathrm{m\,a^{-1}}$ at the grounding line at the 79NG. For both scenarios, the value of $k^2$ is $0.067\,\mathrm{s\,m^{-1}}$. The results for both scenarios are shown in Fig. 9a and c, respectively. Additionally, we show the observed velocities (Fig. 9d, Rignot and Mouginot, 2012) and the PISM surface

velocities (Fig. 9b, Aschwanden et al., 2016). Note that the latter is a PISM model output on a regular grid interpolated to the unstructured ISSM grid.

Velocities computed with the reduced ice overburden pressure are generally too low and do not resemble the structure of the fast flowing branches at all. The result from PISM shows distinct branches for the different glaciers, which display a relatively sharp separation from the surrounding area. Note, that PISM also uses a basal hydrology model as described in Bueler and van

Pelt (2015). Velocities are slightly lower than observed velocities especially for Zacharias Isbrae and in the area, where ZI and 79NG are closest. In the upper part towards the ice divide, the ice stream structure is not visible in the velocities. The ISSM model using effective pressure computed by CUAS produces high velocities towards the ocean that closely resemble $N$. The observed sharp transition between the ice streams and the surrounding ice is poorly reproduced. While the stream structure is way too diffused, the different branches can be discerned and the velocity magnitude for the glaciers appears reasonable. The

inland part is similar to observed velocities but – as in the PISM simulation – the upper part where NEGIS is initiated is not present. The onset of NEGIS is thought to be controlled by high local anomalies in the geothermal flux (Fahnestock et al., 2001), which PISM currently does not account for. Higher geothermal flux would lead to more basal melt, hence, water supply in the hydrology model. However, the consequences for the modeled effective pressure would require further experiments which are not in the scope of this paper. In Tab. 4, we present some statistics of the results: the root mean square error ($L2$-norm), Pearson

correlation coefficient $r^1$, and $\Delta v$ ($L1$-norm) between the modeled and observed velocities.

We find it impressive that even without extensive tuning, we can considerably improve the velocity field in ISSM by our simple one-way coupling to the hydrology model. The results in this section are not to be understood as a thorough study of the NEGIS, but only as a first application of the model to a real geometry. A complete study requires extended observations in order to determine the optimal model parameters. However, we are confident that our results represent the general aspects of

**Table 4.** Comparison of modeling results for horizontal ice velocity to observed values (Rignot and Mouginot, 2012). Herein RMS denotes the root mean square error or $L2$-norm, $r^2$ is the Pearson correlation coefficient, and $\Delta V$ is the $L1$-norm.

|  | RMS $(\mathrm{m\,a^{-1}})$ | $r^2$ | $\Delta v$ $(\mathrm{m\,a^{-1}})$ |
| --- | --- | --- | --- |
| ISSM with reduced ice overburden pressure | 152.30 | 0.77 | 78.63 |
| PISM (Aschwanden et al., 2016) | 132.05 | 0.84 | 65.42 |
| ISSM with $N$ computed from CUAS | 98.62 | 0.90 | 44.39 |

the hydrological system at NEGIS. Based on our sensitivity and seasonal experiments (Sect. 3.1 and Sect. 3.3) we expect the high-transmissivity-areas to be a stable feature, which would extend or retract depending on the chosen values of the melt and creep parametrizations but not change their location. Available supply plays a more important role here, and we assume that different basal melt distributions – or the addition of surface melt – might considerably change the position and the extent of the efficient system and, therefore, the effective pressure distribution as can be seen in Sect. 3.3.

The onset of NEGIS is not well reproduced in the PISM simulation as well as in our ISSM result. Since the ice is slow in the PISM results in that area, basal melt rates are low, and, since we use these as input in our hydrology model, it is expected that our model computes low water pressure here. In our opinion, this represents another point in favor of having a real two-way coupling between the ice model and the basal hydrology model in order to obtain good results. These results could then in turn be used to guide further optimization of the modeling parameters in our hydrology model in the future.

## 5 Conclusions

We present the first equivalent porous medium model for subglacial hydrology that includes the treatment of unconfined water flow. It uses only a single conductive layer with adaptive transmissivity. Since extensive observations of the subglacial system are rare, our approach to fit a simple parametrization of the effective Darcy model to the available data can be an advantage.

We find strong model sensitivity to grid spacing $dx$, the parametrization of melt $a_{\mathrm{melt}}$, creep closure $a_{\mathrm{creep}}$, and the cavity opening parameter, while the sensitivity to the limits of transmissivity and the confined–unconfined transition parameter $d$ is low. Our model robustly reproduces the seasonal cycle with the development and decline of the effective system over the year.

In our NEGIS experiments, we find the presence of a partial efficient system for winter conditions. The distribution of effective pressure broadly agrees with observed velocities, while the upstream part is not represented correctly. When coupled to ISSM, our hydrology model notably improves computed velocities.

A number of aspects of the proposed model can be further developed; those include improved parametrizations of several physical mechanisms (e.g. adding feedback between pressure and water supplies), changing the hydraulic transmissivity coefficient to a tensor-valued to better represent the anisotropy of channel networks, and, last but not least, transition to a mixed formulation of the Darcy equation discretized on an unstructured mesh in order to preserve mass conservation and to improve resolution in the areas of interest.

## Appendix A: Transmissivity evolution details

The channel cross-sectional area $A_c$ expands when there is more melt than ice inflow due to creep, thus the mass change per unit length (unit: $\mathrm{kg\,m^{-1}\,s^{-1}}$) caused by melt ($\dot{M}_\mathrm{melt}$) and creep ($\dot{M}_\mathrm{creep}$) is given as (Cuffey and Paterson, 2010, Eq. 6.42):

$$\rho_i \frac{\partial A_c}{\partial t} = \dot{M}_\mathrm{melt} - \dot{M}_\mathrm{creep}. \tag{A1}$$

5   This is equivalent to

$$\rho_i \frac{\partial b_c}{\partial t} = \dot{m}_\mathrm{melt} - \dot{m}_\mathrm{creep}, \tag{A2}$$

which describes the mass change per unit area (unit: $\mathrm{kg\,m^{-2}\,s^{-1}}$). Note, that $A_c$ is the channel volume per unit length and $b_c$ is the same channel volume, but per unit area and thus a thickness.

Nye (1976), found for the closure of channels due to creep that

$$\frac{1}{R_c} \frac{\partial R_c}{\partial t} = A \left( \frac{N}{n} \right)^n, \tag{A3}$$

with $R_c$ denoting the channel radius (notation as in Cuffey and Paterson (2010, Eq. 6.15)). $A$ and $n$ are the creep parameters for ice given in Table 1 and Table 2. Multiplication by $2\pi\rho_i R_c^2 = 2\rho_i A_c$ on both sides, leads to

$$2\pi\rho_i R_c \frac{\partial R_c}{\partial t} = 2\rho_i A_c A \left( \frac{N}{n} \right)^n \tag{A4}$$

Rewriting the left side to area, using the chain rule ($\partial A_c / \partial t = 2\pi R_c \partial R_c / \partial t$) yields

$$\rho_i \frac{\partial A_c}{\partial t} = 2\rho_i A_c A \left( \frac{N}{n} \right)^n, \tag{A5}$$

thus,

$$\dot{M}_\mathrm{creep} = 2\rho_i A_c A \left( \frac{N}{n} \right)^n, \tag{A6}$$

or again as a change per unit area

$$\dot{m}_\mathrm{creep} = 2\rho_i b_c A \left( \frac{N}{n} \right)^n. \tag{A7}$$

20   Heat produced over the line element $ds$ in unit time is $Q_w G$ and pressure melting point (PMP) effects are $\rho_w Q_w c_w \mathcal{B} \frac{dP_i}{ds}$, which leads to

$$\dot{M}_\mathrm{melt} L_f = \underbrace{Q_w G}_{\text{heat produced}} - \underbrace{\rho_w Q_w c_w B \frac{dP_i}{ds}}_{\text{PMP effect}} \tag{A8}$$

(Cuffey and Paterson, 2010, Eq. 6.16), where $\dot{M}_\mathrm{melt}$ represents the melt rate (mass per unit length of wall in unit time). The magnitude of gradient of the hydraulic potential is given by

25   $$G = |\nabla \phi_h|, \quad \text{where} \quad \phi_h = \rho_w g h. \tag{A9}$$

Neglecting the PMP effects we obtain

$$\dot{M}_{\mathrm{melt}} = \frac{Q_w G}{L_f}. \tag{A10}$$

As before, we can write that as a change per unit area instead:

$$\dot{m}_{\mathrm{melt}} = \frac{Q'_w G}{L_f}, \tag{A11}$$

where $Q'_w = |qb_c|$ is now the flux per unit length and $q = -K\nabla(h)$ this is

$$\dot{m}_{\mathrm{melt}} = \frac{K\nabla(h)b_c\nabla(\rho_w g h)}{L_f}, \tag{A12}$$

which can be rewritten to

$$\dot{m}_{\mathrm{melt}} = \frac{\rho_w g K b_c (\nabla h)^2}{L_f}. \tag{A13}$$

Inserting $\dot{m}_{\mathrm{creep}}$ from Eq. (A7) and $\dot{m}_{\mathrm{melt}}$ from Eq. (A13) into Eq. (A2) and dividing by $\rho_i$ results in

$$\frac{\partial b_c}{\partial t} = \frac{\rho_w g K b_c (\nabla h)^2}{L_f \rho_i} - 2b_c A\left(\frac{N}{n}\right)^n. \tag{A14}$$

Note, that the right-hand side of Eq. (A14) is used in de Fleurian et al. (2016, Eq. (6)) to evolve the equivalent porous layer (EPL) thickness, representing only the channels.

We assume, that changes within the channel network (e.g. the increase/decrease of cross-sectional area of one, some or many channels or just by the variation in the number of channels) can be translated into a large-scale/average change in equivalent transmissivity[1]. Thus, we obtain our evolution equation for the transmissivity by multiplying Eq. A14 with the constant hydraulic conductivity coefficient $K$ of the EPM as

$$\frac{\partial T}{\partial t} = \frac{g\rho_w K T (\nabla h)^2}{L_f \rho_i} - 2AT\left(\frac{N}{n}\right)^n. \tag{A15}$$

The transmissivity evolution could also be applied to model situations when $K$ is varying without any re-formulation.

We also account for cavity opening due to sliding over bedrock bumps in the model using a similar notation as for the channel evolution above. Cavity opening is related to basal sliding speed $\mathbf{v}_b$ and bump geometry though (Werder et al., 2013):

$$\dot{m}_{\mathrm{cavity}} = \rho_i \beta |\mathbf{v}_b|, \tag{A16}$$

where $\beta = b_r/l_r$ depends on the typical height $b_r$ and distance $L_r$ of the bump. Here we use $\beta$ as a model tuning parameter. Cavity opening again translates into a contribution to the transmissivity evolution and we finally obtain

$$\frac{\partial T}{\partial t} = \frac{g\rho_w K T (\nabla h)^2}{L_f \rho_i} - 2AT\left(\frac{N}{n}\right)^n + \beta |\mathbf{v}_b| K. \tag{A17}$$

[1] A precise quantification of the relationship between the geometrical parameters of this channel network and the effective transmissivity of the EPM lies outside of the scope of this work and is likely to be very complex. Here we assume that the changes in the effective transmissivity linearly depend on the melt and creep processes. This assumption serves as well as any to provide a proof-of-concept for our approach, whereas the search for more sophisticated models supported by a crisp line of physical reasoning is certainly a highly interesting topic to be explored in the future work.

## Appendix B: Discretization

We discretize the transient flow equation (Eq. 6) on an equidistant rectangular grid using a Crank-Nicolson scheme. For sake of completeness, we give the equations for a non-equidistant grid here. For the spatial discretization, we use a second-order central difference scheme (e.g., Ferziger and Perić, 2002) leading to the spatial discretization operator for the head $\mathcal{L}_h$:

$$\mathcal{L}_h = T_{i+\frac{1}{2},j} \frac{h_{i+1,j} - h_{i,j}}{(\Delta_f x)_i (\Delta_c x)_i} - T_{i-\frac{1}{2},j} \frac{h_{i,j} - h_{i-1,j}}{(\Delta_b x)_i (\Delta_c x)_i} + T_{i,j+\frac{1}{2}} \frac{h_{i,j+1} - h_{i,j}}{(\Delta_f y)_j (\Delta_c y)_j} - T_{i,j-\frac{1}{2}} \frac{h_{i,j} - h_{i,j-1}}{(\Delta_b 1)_j (\Delta_c y)_j} + Q \tag{B1}$$

where half-grid values of $T$ denote harmonic rather than arithmetic averages computed using Eq. (7), where

$$(\Delta_c x)_k = (x_{k+1} - x_{k-1})/2, \tag{B2}$$

$$(\Delta_f x)_k = x_{k+1} - x_k, \quad \text{and} \tag{B3}$$

$$(\Delta_b x)_k = x_k - x_{k-1} \tag{B4}$$

denote central, forward, and backward differences, respectively. Re-writing this more compactly in compass notation

$$\mathcal{L}_h = d_S h_S + d_W h_W + d_P h_P + d_E h_E + d_N h_N + Q \tag{B5}$$

with

$$d_W = \frac{T_{i-\frac{1}{2},j}}{(\Delta x)_i^2}, \quad d_E = \frac{T_{i+\frac{1}{2},j}}{(\Delta x)_i^2}, \quad d_S = \frac{T_{i,j-\frac{1}{2}}}{(\Delta x)_j^2}, \quad d_N = \frac{T_{i,j+\frac{1}{2}}}{(\Delta x)_j^2},$$

and $\quad d_P = -(d_W + d_E + d_S + d_N).$ \tag{B6}

We use the Crank-Nicolson semi-implicit method for computing our hydraulic head

$$\frac{\Delta h}{\Delta t} = \Theta \mathcal{L}_h(h^{n+1}) + (1 - \Theta) * \mathcal{L}_h(h^n) \tag{B7}$$

(with $\Theta = 0.5$ for Crank-Nicolson) and then update the transmissivity with an explicit Euler step:

$$T^{m+1} = T^m + \Delta t \left( a_{melt}^m + a_{cavity}^m - a_{creep}^m \right), \tag{B8}$$

where we use a combined forward- backward-difference scheme for the discretization of $(\nabla h)^2$ in Eq. (5):

$$(\nabla h)^2 \approx \frac{1}{2} \left[ \left( \frac{h_{i,j} - h_{i-1,j}}{(\Delta_b x)_i} \right)^2 + \left( \frac{h_{i+1,j} - h_{i,j}}{(\Delta_f x)_i} \right)^2 + \left( \frac{h_{i,j} - h_{i,j-1}}{(\Delta_b y)_j} \right)^2 + \left( \frac{h_{i,j+1} - h_{i,j}}{(\Delta_f y)_j} \right)^2 \right]. \tag{B9}$$

Compared to central differences, this stencil is more robust at nodes with large heads caused by moulins.

The time step is chosen sufficiently small so that the discretization error is dominated by the spatial discretization. Additionally, we check that the time step is small enough for the unconfined component of the scheme to become active by restarting the time step with a decreased $\Delta t$ if at any point $h < z_b$.

All variables are co-located on the same grid, but the transmissivity $T$ is evaluated at the midpoints between two grid cells using the harmonic mean due to its better representation of transmissivity jumps (e.g. at no-flow boundaries). A disadvantage of this discrete formulation is that it is not mass-conservative (see, e.g. Celia et al. (1990)). The solution to this is to use a mixed formulation for Darcy flow in which also the Darcy velocity is solved for. However, in our application, the resulting error is very small, and we plan to implement the mixed formulation approach in future work.

*Competing interests.* The authors declare that they have no conflict of interest.

*Acknowledgements.* This work is part of the GreenRISE project, a project funded by Leibniz-Gemeinschaft: WGL Pakt für Forschung SAW-2014-PIK-1. We kindly acknowledge the efforts of Basile de Fleurian und Mauro Werder who were designing and supporting the SHMIP project. We highly benefited from their well developed test geometries and fruitful discussions not only at splinter meetings. Basile helped in the development of the method by suggesting unconfined aquifer flow as a solution for negative water pressure.

We acknowledge A. Aschwanden for providing basal melt rates, temperatures and velocities simulated by PISM. Development of PISM is supported by NASA grants NNX13AM16G, NNX16AQ40G, NNH16ZDA001N and by NSF grants PLR-1644277 and PLR-1603799.

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

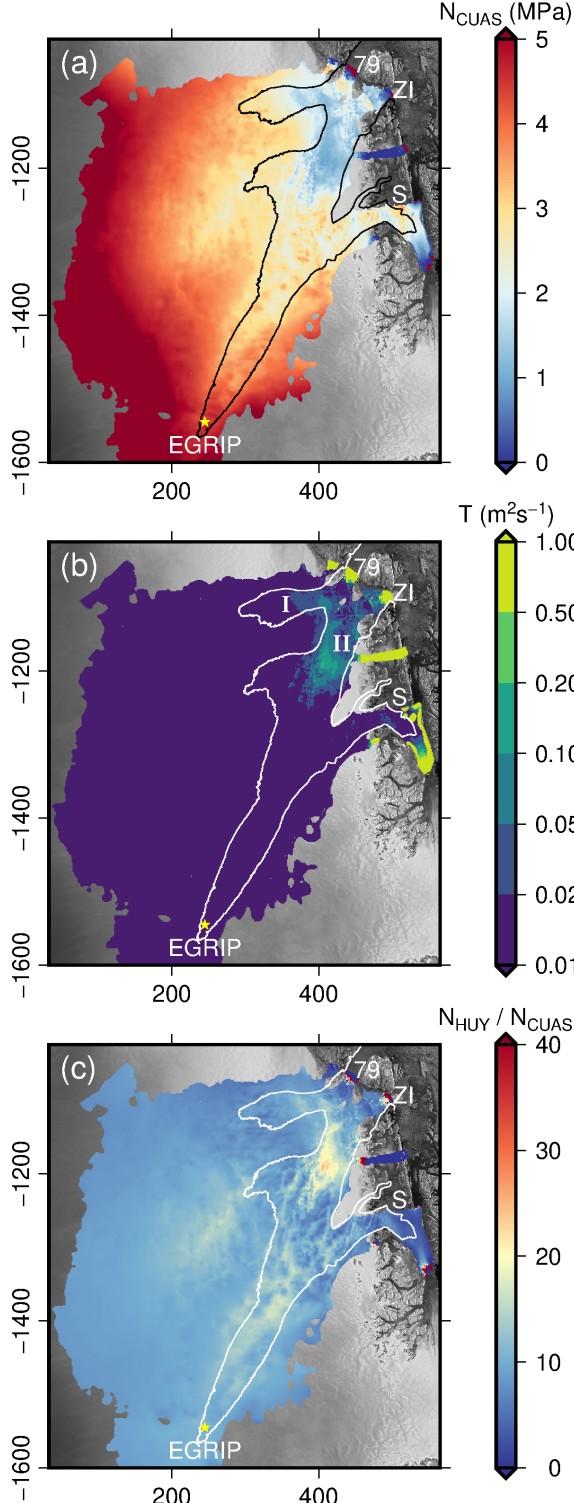

**Figure 8.** Results for NEGIS region with forcing due to basal melt (PISM) representing winter conditions. White lines indicate the $50\,\text{ma}^{-1}$ velocity contour. Panel (a) shows effective pressure $N_{\text{CUAS}}$, (b) transmissivity $T$ (logarithmic scale), and (c) shows the quotient of the ice overburden pressure above flotation and the effective pressure computed by CUAS.

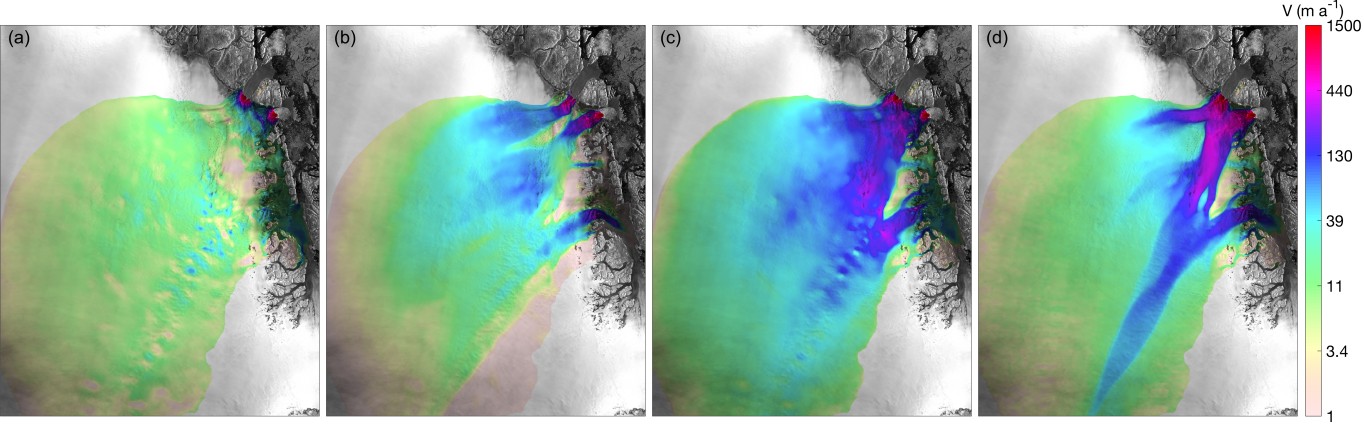

**Figure 9.** Horizontal surface velocity: ISSM with reduced ice overburden pressure $N_{\mathrm{HUY}}$ (a), PISM result from Aschwanden et al. (2016), interpolated to unstructured ISSM grid (b), ISSM with effective pressure from our hydrology model $N_{\mathrm{CUAS}}$ (c), and observed velocities (Rignot and Mouginot, 2012) (d).