# Peer review of "A confined–unconfined aquifer model for subglacial hydrology and its application to the North East Greenland Ice Stream"

_The Cryosphere, 2017_

## Referee Comment (RC1) · D. Brinkerhoff (Referee) · 30 Nov 2017

**General Comments**

In 'A confined-unconfined aquifer model for subglacial hydrology and its application to the North East Greenland Ice Stream', the authors develop a new model of subglacial hydrology based upon the idea that subglacial water flow can be approximated as a layer of variably permeable material (a linked cavity system or an actual till aquifer) topped with a mostly inpermable cap (the glacier). The paper makes the claim that this approach has lower computational requirements than more explicit models and can

simulate effective pressures suitable for forcing sliding laws in ice sheet models. Several examples illustrate model results, both synthetic (SHMIP) and inspired by reality (NEGIS).

**Specific Comments**

Model formulation

I appreciate the novel thinking showcased in this paper; it's always useful to approach an old problem in a new way, and by switching from the classical model formulation used in most contemporary ice sheet models to this new viewpoint, the authors certainly introduce a new way of thinking. Unfortunately, I think that this viewpoint lacks physical justification. The primary way in which this paper departs from previous subglacial hydrology modelling efforts is that differences in flux are accounted for by changes in the conductivity, rather than a change in the average cavity size. However, the equations for evolving cavity size (which are well understood from a theoretical perspective) are used to model the change in conductivity, with units made to match by simply multiplying by the conductivity. How is this justified? Without any theoretical justification, the model becomes strictly heuristic, and if this is the case, why is this model formulation any better than any other random model formulation that happens to achieve results that compare favorably to SHMIP? If the authors can provide such physical justification, I will happily withdraw this criticism. If not, then I want to see this point stated prominently in the paper.

There is a sign error in Equation 9: the creep term should be negative. Also, $v_{melt}$ as written implies that melt is always based on a fully saturated aquifer. The $b$ in that term should be replaced with $\min(b, h)$.

Interactive
comment
Coupling to basal sliding

The chosen formulation neglects the coupling between sliding and hydrology. Most models of subglacial hydrology allow for the opening of cavities (and hence an increase in the effective transmissivity) by accounting for ice cavitation over sub-grid scale bedrock asperities. In contrast, this paper assumes that transmissivity (what I can only view as a proxy for the opening of cavities or channels) opens only by dissipative melting. This is problematic for several reasons. First, if the authors don't think that this is an important mechanism in making space for water to move around in below a glacier, then they need to say so. Essentially all work on this subject recognizes this as a major process, particularly in cases with significant sliding. Second, it is fairly well understood that in the continuum approximation, when this term is dominant, the problem always leads to runaway channelization, precluding the presence of linked cavities (hence the use of edge-based formulations for modelling channels in e.g. Werder (2013)). Why is this not a problem here, in both a numerical and a physical sense? Finally, the paper includes an extra parameters $K_{min}$, which makes it so that there is always transport capacity, and this is identified as a sensitive parameter. This parameter strikes me as a hack to solve a problem that would be solved in a more principled way by including a term that increases conductivity proportionally to sliding speed.

Transmissivity formulation

The principle variable entering the mass conservation formulation is $h$, defined as the piezometric head, or the potential relative to some fixed datum. This is fine, so long as the appropriate modifications are made when the head drops below the bedrock elevation (this does not seem to be handled at all in this model). However, the transmissivity $T(h)$ is formulated as if $h$ were the height above bedrock. Either the transmissivity

should be

$$T(h) = \{ \, K \, (b - B), h \geq b; \, K(h - B), B < h < b; \, 0, h < B \tag{1}$$

and $b$ should be redefined as the aquifer thickness plus bedrock elevation, or $h$ should be redefined as the local height above bedrock and the mass conservation equation should be changed to

$$S_e h_t = \nabla \cdot T(h) \nabla (B + h) + Q. \tag{2}$$

Note that this error makes no difference when the bedrock elevation is uniformly zero (SHMIP, for example). However, it would lead to some very questionable results when there are significant variations.

Effective storage coefficient

Why does $b$ appear in $S_e(h)$? if $h > b$, then the head is rising through glacier ice with permeability $S_s$ (this is regularization to make the equation parabolic rather than elliptic, see Schoof (2012)). Why then would the head increase depend at all upon the thickness of the underlying aquifer? It is also worth noting that $S_s$ as presented has units that don't make sense (Pa$^{-1}$ m$^{-1}$), and that the porosity $\omega$ cancels out in Eq. 2. I also think that there is a misunderstanding with respect to the meaning of the effective storage coefficient. This is simply the void space in the aquifer versus in the glacier: it makes sense to say that the aquifer is more porous than the glacier (i.e. the head changes faster in a confined aquifer than an unconfined one), but not that more water is released from an unconfined aquifer than a confined one.

Discretization

Given that you're using central differences and forward Euler to discretize (not exactly revolutionary), this section could be moved to an appendix or supplement, or even omitted all together.

What cannot be omitted is a discussion of stability under time stepping. In particular, the Courant-Friedrich-Lewy criterion imposes a time-step restriction for stability in the case of explicit time steps. For the chosen discretization, it is very small indeed. I would like to see some verification that the CFL is being respected. I suspect that it is currently not, which would provide a potential explanation for the obvious oscillations (i.e. checkerboard pattern) that appear in the results.

Also, how bad is the mass conservation problem? If a considerable amount of mass is being lost or gained, then this affects the validity of the results.

Parameter choices

The choices of $\omega$ and $b$ are incompatible in most plausible scenarios for a glacier base. In the case of a sediment aquifer, then a value of $b = 10$ m is reasonable, but $\omega = 0.4$ (40% void space!) is not at all reasonable. Conversely, if the 'aquifer' is the linked cavity/conduit system, then $\omega = 0.4$ might be reasonable but $b = 10$ m is too large by an order of magnitude. A much stronger effort needs to be made to state the type of physical system that the model is supposed to simulate, and parameter values with regards to this need to be better justified.

Figure 3j

Is there any transport between channels? It doesn't seem like it from this figure. Shouldn't efficient channels reduce the pressure, causing water to flow in laterally, eventually leading proximal channels to merge?
$N_{HUY}$

The use of 'reduced ice overburden pressure' as a comparison in this case is a bit of a straw man. As it appears in Huybrechts (1990), this refers to a water pressure given by bedrock elevation below sea level, which is only reasonable when very little basal or surface melt is expected (as in the Antarctic context for which it was initially used). For NEGIS, a much more defensible comparison would be that the water pressure is bounded below by sea level height, but that otherwise it is a constant fraction of overburden. Also, isn't $N_{CUAS}^{-1}/N_{HUY}^{-1} = N_{HUY}/N_{CUAS}$? Why the exponents?

On the definition of 'improvement'

What constitutes an improvement per 'we can considerably improve the velocity field in ISSM ...'? An improvement with respect to the eyeball norm? Or is it possible to be somewhat more quantitative, e.g. computing the misfit between these results, PISM, and ISSM without the hydrological model?

**Technical corrections**

**P1L16** Citation needed

**P2L1–3** Citations needed

**P2L24** How do you know that this strategy captures the overall behavior?

**P2L30** Should cite Schoof (2012) or Bueler (2015) or some other paper that discusses the implications of assuming that the system is always full

**P3L20** This statement is not true in the subglacial hydrology literature

**P16L16** Citation needed

**P16L32** Should be 'bed model of Morlighem (2014)' rather than 'data'. The results of a PDE-constrained optimization scheme are not data

**P19L14** What does 'empirical nature' mean?

**P19L17** Not sure that 'restitution' is the right word.

---

## Referee Comment (RC2) · Anonymous Referee #2 · 7 Dec 2017

This paper presents a new model of subglacial hydrology, based on concepts from groundwater hydrology. A distributed system is modelled as a confined/unconfined till aquifer, while the effects of channelization are mimicked by varying the permeability/storage of the aquifer. This approach to subglacial hydrology is similar in nature to De Fleurian et al. (2014);  however, rather than modelling channels as a separate layer, this approach allows the subglacial system to be modelled using a single layer. The aim of this approach is broadly capture the behavior of the subglacial system, rather than to model the impact of discrete channels. This upside is thats its designed to be computationally cheaper than more explicit models such as Werder et al., (2012) or Hewitt (2013).  The authors show the behavior of the model on synthetic test cases from SHMIP, and present a real world application to NEGIS.

Efforts to parameterize the subglacial system in a simple, computationally efficient manner are important for incorporating subglacial hydrology into modelling studies covering large spatial and temporal scales. The work of the authors is novel and addresses an important area of research. However, I have several concerns that the authors should to address.

Assumption of till

I think an implicit assumption in the model is a soft-bed subglacial hydrology system. This should be clarified. Since the model is applied to Greenland, I think it would be relevant to briefly cover literature discussing whether a soft-bed system exists there, since there is disagreement about the nature of the bed. This could include recent seismic studies, and touch upon the previous modelling studies making this assumption (e.g. Bougamont et al, (2014)). Further, it may also be beneficial to readers to discuss the relationship between till and sliding laws.

Model Formulation

The model description is unclear about how to conceptually understand this model. I am puzzled about the motivation and physical interpretation of scaling K with channel opening/closing while the aquifer is unconfined. This would imply that channels are forming in the porous medium? The equations for channels are not applicable there. I'm perplexed since I would expect K to be constant (or perhaps depend on other variables like strain rate, stress, sediment properties) in the unconfined case, and then force a switch to channelized behavior once the aquifer becomes confined.

In your model formulation, you scale conductivity (Eq 9) using the equations for channel opening/closing. However, wouldn't it be more appropriate for K be scaled such that flux through a grid cell scales to flux through an idealized channelized system? In other words, I would expect an attempt to scale $\nabla \cdot (T(h)\nabla h)$ with discharge through channels (using an assumed channel spacing). Where the discharge through channels is (e.g. Equation 3 from Hewitt (2013))

$$Q = -K_c S^{(5/4)} |\frac{\partial \varphi}{\partial s}|^{(\frac{-1}{2})} \frac{\partial \varphi}{\partial s}$$

When channelization is introduced into models, they can grow unstably and dominate the system as effective pressures in channels decreases with increasing input. This doesn't appear to occur in your model. You should give a physical description of the terms (and point out the terms in your equations) preventing this.

The argument that the amount of water released from an unconfined aquifer is larger than a confined aquifer is counterintuitive (P4L5). If the head drops 1m in the unconfined case, than wouldn't the water released be much greater than in the confined case, due to the volume of water in the latter being limited by the porosity?

Numerics

Your conductivity doesn't appear to show grid convergence, even at resolution of 500m. However, I would expect large scale model runs to require convergence at much coarser resolutions. While you identify the conditions under which you get the checkerboard pattern, you don't really explain whats causing it to form. Is this not an artificial pattern due to the numerics? You're solving a highly non-linear equation. In a second order discretization, the dominant truncation error is odd (third order), and hence the error generally will behave in a dispersive manner. Equations 7 and 8 suggest that $N \sim \Pi_i - h$. Since N appears smooth in your plots, this implies h should be smooth, and then I'm uncertain why K isn't smooth as well.

Application to NEGIS

Your study domain encompasses areas of both fast flow and slow flow. While the assumptions of the SSA are valid for the ice stream itself, the SSA is not the right approximation for ice flow over the majority of the domain. This is evident in Fig 10a, where we can see that your modelled ice speed is ~1 m/a over the majority of the domain. Over this part of the domain, internal deformation is the key component of ice flow. The comparison of panels A and B in Figure 10 shows not only the effect of subglacial hydrology, but also of different ice physics. Aschwanden et al. (2016) use a 'hybrid' model described in Bueler and Brown (2009), which is itself an approximation (in effect) to the Blatter-Pattyn approximation. It is worth noting that other 'hybrid' approximations exist, such as L1L2 or that of Goldberg (2011). Because of the different regimes in your domain, I think it is necessary to either use a hybrid approximation (if its available in ISSM), or Blatter-Pattyn to test the results of your coupled model.

Minor Comments:

P1L2: ...'drives freshwater into the ocean'... State the explicit impact of this (e.g. undercutting at calving fronts?)

P1L13-16 This paragraph would benefit from references.

P2L1: 'predominant in alpine glaciers and on the margins of the Greenland': This has more nuance, as channels develop seasonally, and are not predominant year round.

P2L7: I think Hewitt (2013) should be mentioned here, as should Hoffman and Price (2014)

P2L8: …'remarkable results for spontaneously evolving channel networks'. The wording/concept could use clarification

P2L27-28: 'While the assumption … with lower water input'. Citation needed

P3L15: hydraulic head needs a definition

P3L18: Eq 2: porosity cancels itself out. Can you confirm that the units match up?

P3L19: the definition of alpha in table 1 reads the 'compressibility of water'. The definitions of alpha/beta_w in table 1 needs to be switched I think.

P3L25: although h is defined as the hydraulic head, it appears to be used as the saturated height

P3L28: In Equation 5, it is unclear why Se(h) in the unconfined case depends on b. When the aquifer is unconfined (say with a saturated thickness of 1m), does it matter if the aquifer thickness is 10m or 20m? To gain a better overview of this formulation, I looked at the 'Groundwater flow equation' page on wikipedia (https://en.wikipedia.org/wiki/Groundwater_flow_equation). Although I admit that it is not an authoritative source, the formulation there states that Se(h) = S_s*b in the confined case, and S_e(h) = S'(h) is the unconfined case.

P4Fig1: This should be updated/supplemented to show the physical interpretation of the models with channels.

P5L17: I would move discretization to an appendix. It's beneficial to any reader looking to reimplement your model, but not necessary in the main text.

P7L16: Can you cite the upcoming results as Author(s) (In Prep)?

P8 Table 3: I think it would be beneficial to discuss these values compared to the inferred hydraulic conductivity values of till: 10^-9 - > 10^-4 m/s (Fountain and Walder, 1998).

P11L4: 'less' → 'lower'

P13L18: basal topography has no influence at all in the unconfined case?

P17L19: You need to define N_HUY.

P18L1: the quotient of X^-1 and Y^-1 could be simplified to the quotient of Y and X.

P18L8: It's important to add a citation here to MacAyeal (1989) and/or Morland (1987). In particular, the term SStA is confusing, since this approximation is often known as the SSA in the community. There is a proliferation of 'hybrid' models now, combining SSA and SIA, so a variation like SStA could be misinterpreted as one of those. I looked up the ISSM documentation to be sure that SStA was equivalent to SSA.

P18L31: The improvement/comparison of your results should be quantified.

P19L8. 'This decisively illustrates the importance of having a real two-way coupling between the ice model and the basal hydrology model in order to obtain good results.' I don't believe you have shown this. You would have to show that you cannot reproduce the results with 2-way coupling, and then show that you reproduce the results when 2-way coupling is introduced. Since it is not cited, I would point the authors to Hoffman and Price (2014) for a detailed discussion on coupling of ice flow and subglacial hydrology.

P19L17: This sentence reads oddly.

P24Fig10. The descriptions and panel labels are mixed up [there is no (d) in the figure caption].

---

## Author Comment (AC1) · 22 Dec 2017

Dear Editor, dear Reviewers,

the authors wish to thank both reviewers for careful reading of the manuscript, helpful remarks and constructive criticism. We want to precede this response by a somewhat general explanation of our modeling concept that appears to have caused a great deal of misunderstanding -- likely due to somewhat unspecific model motivation in our paper: Our goal is not to fit an aquifer model to the really existing subglacial aquifers by finding correct geometrical and physical parameters. This approach would likely work for flows through till or similar types of drainage but just as likely fail for channel networks. Instead we chose Darcy model as an 'idealized' representation of all drainage processes (including channels) and tuned it to give the best fit to the available data. This, in particular, means that channels are not being explicitly modeled (and hence no channel equations are being used) but, instead, the effect of channel drainage is represented by appropriately chosen hydraulic conductivity coefficients in the framework of the Darcy equation. To better explain this point, we added a corresponding sentence to the introduction.

We also want to clarify that not representing cavities in our model does not mean that we not aware of numerous sophisticated studies on cavity formation that appeared in the past decades, but rather state that our focus is the channels that clearly top our priority list -- partly due to our studies of 79NG in which we have been looking for a long time for locations where field studies could potentially survey subglacial channels.

In general, many points raised in the reviews, are highlighting that (more) observations of subglacial hydrology underneath ice sheets are urgently required.

Technicalities: below we answer each point raised by the reviewers and mark our answer in blue color, whereas the original comment of the reviewer is shown in black. Point raised by both reviewers are answered at one location and referenced at the second one.

We have performed the following major changes - that are all also documented below in detail:
- Appendix A - presents the derivation of the channel opening and closure terms
- We improved the discretisation of v_melt term to prevent occurrence of checkerboard patterns and rerun all simulation
- In Eq 10, we changed b to min(b,h-z_b) as suggested by the reviewer
- We moved the numerical discretisation description into the Appendix B

**First review**

D. Brinkerhoff

**Specific Comments**
**Model formulation**
I appreciate the novel thinking showcased in this paper; it's always useful to approach an old problem in a new way, and by switching from the classical model formulation used in most contemporary ice sheet models to this new viewpoint, the authors certainly introduce a new way of thinking.

1) Unfortunately, I think that this viewpoint lacks physical justification. The primary way in which this paper departs from previous subglacial hydrology modelling efforts is that differences in flux are accounted for by changes in the conductivity, rather than a change in the average cavity size. However, the equations for evolving cavity size (which are well understood from a theoretical perspective) are used to model the change in conductivity, with units made to match by simply multiplying by the conductivity. How is this justified? Without any theoretical justification, the model becomes strictly heuristic, and if this is the case, why is this model formulation any better than any other random model formulation that happens to achieve results that compare favorably to SHMIP? If the authors can provide such physical justification, I will happily withdraw this criticism. If not, then I want to see this point stated prominently in the paper.

**In order to address the concerns about a our model being heuristic, we provide the derivation of Eq. 9: The key parameter describing the efficiency of drainage is the transmissivity that is computed as the product of the conductivity with the effective layer thickness (Eq. 4). Thus, mathematically speaking, it is fully equivalent whether we model the conductivity with fixed thickness (as in our study) or other way around (as was done in de Fleurian et al. 2016). To supplement this, we added Appendix A to the revised manuscript describing the derivation of the melt and creep terms.**
**To sum it up, in terms of physical foundation, we generally stay very close to modeling approaches used in de Fleurian et al. 2016 with the main difference being not the representation of cavity size /effective conductivity relationship, but the use of a single Darcy layer capturing both drainage systems instead of relying on two distinct layers with rather complex coupling conditions between them.**

2) There is a sign error in Equation 9: the creep term should be negative.
**Thank you for pointing it out. This was just a typo in the manuscript, the code is using the correct sign. We have changed it accordingly in Eq. 9.**

3) Also, $v_{melt}$ as written implies that melt is always based on a fully saturated aquifer. The $b$ in that term should be replaced with $min(b, h)$.

**This is correct, we did not account for that until now. We have updated this in our model and re-run the experiments. However, the impact of this change in our setups was negligible.**

4) Coupling to basal sliding

The chosen formulation neglects the coupling between sliding and hydrology. Most models of subglacial hydrology allow for the opening of cavities (and hence an increase in the effective transmissivity) by accounting for ice cavitation over sub-grid scale bedrock asperities. In contrast, this paper assumes that transmissivity (what I can only view as a proxy for the opening of cavities or channels) opens only by dissipative melting. This is problematic for several reasons. First, if the authors don't think that this is an important mechanism in making space for water to move around in below a glacier, then they need to say so. Essentially all work on this subject recognizes this as a major process, particularly in cases with significant sliding. Second, it is fairly well understood that in the continuum approximation, when this term is dominant, the problem always leads to runaway channelization, precluding the presence of linked cavities (hence the use of edge-based formulations for modelling channels in e.g. Werder (2013)). Why is this not a problem here, in both a numerical and a physical sense? Finally, the paper includes an extra parameters K min, which makes it so that there is always transport capacity, and this is identified as a sensitive parameter. This parameter strikes me as a hack to solve a problem that would be solved in a more principled way by including a term that increases conductivity proportionally to sliding Speed.

**Indeed, our model does so far not include any opening of cavities, which we plan as one of the next extensions of the model. If we incorporate cavity opening, we will, however, face the challenge that the size of undulations is not known from observations at the resolution that is required. The basal roughness is typically measured with radio echo sounding and hence, depending on the instrument, one may end up with vertical resolution between 5m and 50m for airborne applications. So, once we incorporate this for artificial geometries into our model, we will lack the observations of basal undulations for real world applications. Here we would need to come with some assumptions on roughness again, which then not too much different from using a K_min that only mimics a transmission through cavities. So in some way, we account for this by having a K_min, which does allow a minimum transmission of water.**

**Fowler (1987) discussed the potential for cavitation in ice sheets and glaciers and found that cavitation is ruled out for large ice sheets, due to the slope of ice sheets and the length scale of bed undulations. This is likely to be valid for the major part of the ice sheets, while at the margins situation might become closer to valley glaciers in terms of surface slope. The NEGIS itself exhibits, however, also quite some surface undulations, so a cavity model would be definitely interesting to apply to this area.**

**One could, on the other hand, also argue that there are no direct observations of subglacial channels in Greenland and question, based on that, their role as a key component of the hydrological system. Indeed, the only observation we are aware for the surface representation of a subglacial channel drainage might be p. 13 of http://old.esaconferencebureau.com/Custom/14C19/Presentations/02%20day%202%20(Wednesday,%2017%20September)/Session%207/1440_Nagler.pdf giving an indication of width and size of subglacial channels without being a real in-situ observation of a subglacial channel. Existence of subglacial channels is on the other hand also consistent with the observation of plumes in fjords at tidewater glaciers and would also be consistent with findings of melt channels underneath the floating tongue glaciers, e.g. Petermann and 79NG. Thus we were targeting channels rather than cavities in our model first.**

**We discuss the issue of runaway instability further below (see 4) in the second review).**

**Bindschadler 1983, ''The Importance of Pressurized Subglacial Water in Separation and Sliding at the Glacier Bed, Journal of Glaciology, Vol 29, No 191, p. 3-19**
**Fowler 1987, "Sliding with cavity formation" Journal of Glaciology, Vol 33, No 115, p. 255-267**

5) Transmissivity formulation
The principle variable entering the mass conservation formulation is h, defined as the piezometric head, or the potential relative to some fixed datum. This is fine, so long as the appropriate modifications are made when the head drops below the bedrock elevation (this does not seem to be handled at all in this model).
**We do handle this problem via the unconfined aquifer formulation and illustrate this in Section 3.2: without the unconfined formulation, the head drops below the bedrock, which leads to negative water pressure. When considering the transition to unconfined aquifer, as soon as the head drops below the top of the aquifer, the transmissivity is reduced, reaching zero when the head reaches the bedrock. This prevents it from dropping below the bedrock. This may have been unclear because of our erroneous formulation of transmissivity (see next comment), and we added a note to make this clearer:**
*"This also prevents the head from falling below the bedrock, as we detail in Section 3.2."*

6) However, the transmissivity $t(h)$ is formulated as if h were the height above bedrock. Either the transmissivity should be $T(h) = \{K(b-B), \ h \geq b; \ K(h-B), \ B < h < b; \ 0, \ h < B$ and $b$ should be redefined as the aquifer thickness plus bedrock elevation, or $h$ should be redefined as the local height above bedrock and the mass conservation equation should be changed to $S_e h_t = \nabla \cdot T(h) \nabla (B + h) + Q$
Note that this error makes no difference when the bedrock elevation is uniformly zero

(SHMIP, for example). However, it would lead to some very questionable results when there are significant variations.

**Thank you for pointing this out. We had it correctly in the code (same formula as the first suggestion) and now corrected the formulation in the paper by replacing h by $\Psi = h - z_b$ defined as the height of the head above bedrock.**

7) Effective storage coefficient

Why does $b$ appear in $S_e(h)$? If $h > b$, then the head is rising through glacier ice with permeability $S_s$ (this is regularization to make the equation parabolic rather than elliptic, see Schoof (2012)). Why then would the head increase depend at all upon the thickness of the underlying aquifer?

**$S_s$ is not the permeability of the glacier ice in our model but rather the specific storage of the aquifer under the glacier that we use as an idealized model of the drainage system. In this idealized model, the effective behavior in the confined case is modeled according to the aquifer equations (and thus depends on b). Just as pointed out by the reviewer, it can also be considered as a small regularization parameter adding some elasticity to the model in the confined case. Also see our answer to 9) below.**

8) It is also worth noting that $S_s$ as presented has units that don't make sense (Pa$^{-1}$m$^{-1}$), and that the porosity ω cancels out in Eq. 2.

**There was a typo in Eq. 2, missing a '+'. It should read $\rho_w \omega\, g\, (\beta_w + \frac{a}{w})$.**

9) I also think that there is a misunderstanding with respect to the meaning of the effective storage coefficient. This is simply the void space in the aquifer versus in the glacier: it makes sense to say that the aquifer is more porous than the glacier (i.e. the head changes faster in a confined aquifer than an unconfined one), but not that more water is released from an unconfined aquifer than a confined one.

**We agree that 'released' is not really a good term to describe this term and changed the formulations in the paper accordingly. However, the idea of glacier 'storing' water is not connected in any way with our modeling assumptions (or with our interpretation of $S_s$). The idealized model represents the main effective properties of the *entire* system that are then modeled using aquifer equations.**

10) Discretization

Given that you're using central differences and forward Euler to discretize (not exactly revolutionary), this section could be moved to an appendix or supplement, or even omitted altogether.

**We moved the section to the Appendix B.**

11) What cannot be omitted is a discussion of stability under time stepping. In particular, the Courant-Friedrich-Lewy criterion imposes a time-step restriction for stability in the case of explicit time steps. For the chosen discretization, it is very small indeed. I would like to see some verification that the CFL is being respected. I suspect that it is currently not, which would provide a potential explanation for the obvious oscillations (i.e. checkerboard pattern) that appear in the results.

**We do compute our time steps so that the CFL-criterion is always respected and plan to improve the model by using an implicit time-stepping method in the future.**

**We have now found the cause of the oscillations: The flux that governs the melt term is computed in the center of the cell, while the correct position would be at the boundary between cells; this lead to the observed instability. Once we changed the stencil for $v_{melt}$ accordingly, the instabilities went away. The new results have slightly smoother distribution of $K$, but the general behaviour is the same as before.**

**We re-run all simulations with the new formulation and updated our paper accordingly.**

12) Also, how bad is the mass conservation problem? If a considerable amount of mass is being lost or gained, then this affects the validity of the results.

**The error in mass conservation is very small. Below is a figure for the evolution of water mass in the system for the SHMIP experiment (using our default parameters), where $W$ is the water that is stored in the aquifer.**

[Figure]

13) Parameter choices

The choices of ω and b are incompatible in most plausible scenarios for a glacier base.

In the case of a sediment aquifer, then a value of $b$ = 10 m is reasonable, but ω = 0.4 (40% void space!) is not at all reasonable. Conversely, if the 'aquifer' is the linked cavity/conduit system, then ω = 0.4 might be reasonable but $b$ = 10 m is too large by an order of magnitude. A much stronger effort needs to be made to state the type of physical system that the model is supposed to simulate, and parameter values with regards to this need to be better justified.

**As already mentioned before, our model parameters (including aquifer geometry) are not directly related to the measurements of an actual subglacial aquifer but rather represent an 'idealized' aquifer that would display characteristics similar to the drainage system we try to simulate. Since this drainage system includes both types of transport mechanisms, the chosen parameters reflect this duality.**

14) Figure 3j

Is there any transport between channels? It doesn't seem like it from this figure. Shouldn't efficient channels reduce the pressure, causing water to flow in laterally, eventually leading proximal channels to merge?
**There is some transport between the areas of high conductivity seen in Fig. 3j (we do not model channels explicitly). However, these areas do not merge in this experiment probably because of the strong forcing of constant moulin supply and flat bedrock. Note that in the updated results, the areas of high drainage are more spread out.**

15) $N_{HUY}$

The use of 'reduced ice overburden pressure' as a comparison in this case is a bit of a straw man. As it appears in Huybrechts (1990), this refers to a water pressure given by bedrock elevation below sea level, which is only reasonable when very little basal or surface melt is expected (as in the Antarctic context for which it was initially used). For NEGIS, a much more defensible comparison would be that the water pressure is bounded below by sea level height, but that otherwise it is a constant fraction of overburden.
**We agree that there are other appropriate sliding parameterisations that could be used for a comparison; however, our intention was to select one of those that survived over a long time period and is still present in models participating in initMIP Greenland in order to allow the colleagues to assess how their own parameterisation is benchmarking against the application of CUAS and if using CUAS would be beneficial for their models. For this purpose, we would prefer to keep the comparison to Huybrechts approach; however, we are flexible and would remove it on advice from the editor.**

16) Also, isn't $N^{-1}_{CUAS} / N^{-1}_{HUY} = N_{HUY} / N_{CUAS}$? Why the exponents?
**We chose this description because sliding laws depend on 1/N, and we wanted to show where CUAS leads to more sliding, but we agree that the suggested changes make the presentation clearer.**

17) On the definition of 'improvement'

What constitutes an improvement per 'we can considerably improve the velocity field in ISSM ...'? An improvement with respect to the eyeball norm? Or is it possible to be

somewhat more quantitative, e.g. computing the misfit between these results, PISM, and ISSM without the hydrological model?

**Indeed the eyeball norm is not quantitative enough for this comparison. Here, we present the mean ($\triangle$ v = mean(abs(v_mod-v_obs))), the Pearson correlation coefficient, as well as the l2 norm (RMS) of the difference between simulated and observed horizontal velocities for all three cases presented in the manuscript:**

| Experiment | RMS (m/a) | $r^2$ (Pearson) | $\triangle$ v ($l1$ norm) (m/a) |
|---|---|---|---|
| **ISSM Huybrechts** | **176.83** | **0.69** | **90.13** |
| **PISM** | **132.05** | **0.84** | **65.42** |
| **ISSM CUAS** | **126.86** | **0.80** | **53.28** |

**We incorporated this also in the manuscript.**

**Technical corrections**

**P1L16** Citation needed
**We incorporated the following references:**
**Lliboutry, 1968; Röthlisberger, 1972**
**Gimbert, F., V. C. Tsai, J. M. Amundson, T. C. Bartholomaus, and J. I. Walter (2016), Subseasonal changes observed in subglacial channel pressure, size, and sediment transport, Geophys. Res. Lett., 43, doi:10.1002/2016GL068337.**

**P2L1−3** Citations needed
**We added the following references:**
**Aschwanden 2016**
**Van den Broeke et al. 2017**
**P2L24** How do you know that this strategy captures the overall behavior?
**We feel that our efforts to compare our results to the available benchmarks and indirect measurements (surface velocities of ISSM) (Sections 3 and 4) do justify this claim.**
**P2L30** Should cite Schoof (2012) or Bueler (2015) or some other paper that discusses the implications of assuming that the system is always full
**We add "This problem has been analyzed in detail by Schoof (2012), but here we study the effect in the context of equivalent aquifer models using unconfined flow as a possible solution." after the sentence.**
**P3L20** This statement is not true in the subglacial hydrology literature

**We deleted this statement.**

**P16L16** Citation needed

*The role of subglacial hydrology in the genesis of ice streams in general is not well understood yet.*

**We did not find an appropriate reference for this - papers are either proposing subglacial lakes playing a major role (e.g. Bell et al. 2007, Fricker et al., 2014), while the studies testing this hypothesis in depth are currently under review or white papers, e.g. the proposal for EGRIP, stating it as we do in this manuscript. In case the editor or the reviewers have a suggestion, we will incorporate this.**

**P16L32** Should be 'bed model of Morlighem (2014)' rather than 'data'. The results of a PDE-constrained optimization scheme are not data

**Done.**

**P19L14** What does 'empirical nature' mean?

**We replaced this by "its relative simplicity and representation of the main 'bulk' properties of the drainage system with a small number of tunable parameters can be an advantage.**

**P19L17** Not sure that 'restitution' is the right word.

**Changed to 'decline'.**

**Second review**

Efforts to parameterize the subglacial system in a simple, computationally efficient manner are important for incorporating subglacial hydrology into modelling studies covering large spatial and temporal scales. The work of the authors is novel and addresses an important area of research.

However, I have several concerns that the authors should to address.

1) Assumption of till

I think an implicit assumption in the model is a soft-bed subglacial hydrology system. This should be clarified. Since the model is applied to Greenland, I think it would be relevant to briefly cover literature discussing whether a soft-bed system exists there, since there is disagreement about the nature of the bed. This could include recent seismic studies, and touch upon the previous modelling studies making this assumption (e.g. Bougamont et al, (2014)). Further, it may also be beneficial to readers to discuss the relationship between till and sliding laws.
**We do not assume any type of subglacial system in particular but aim to develop a general parametrization for different types of systems in a simplified way. For the sliding law discussion, also see 4) in the first review.**

2) Model Formulation

The model description is unclear about how to conceptually understand this model. I am puzzled about the motivation and physical interpretation of scaling K with channel opening/closing while the aquifer is unconfined. This would imply that channels are forming in the porous medium? The equations for channels are not applicable there.
**The interpretation of K (and its dependence on the channel opening/closure) does not differ between the confined and unconfined cases. No channels are explicitly formed, instead the conductivity is adjusted to account for the effect of both drainage systems (and, yes, we do allow the efficient drainage system to form in the unconfined case). We added some more specific conceptual explanations for our modeling approach to the introduction (also see the general comments at the beginning of this document).**

3) I'm perplexed since I would expect K to be constant (or perhaps depend on other variables like strain rate, stress, sediment properties) in the unconfined case, and then force a switch to channelized behavior once the aquifer becomes confined.
**K is not the intrinsic permeability but hydraulic conductivity. The former does, in fact, depend on the solid and fluid parameters. The latter is an 'effective' parameter that describes the current ability of the porous medium to conduct fluid (this includes the channels, etc.). The behavior of K that accounts for melt/creep is also assumed to be present in the unconfined case.**

4) In your model formulation, you scale conductivity (Eq 9) using the equations for channel opening/closing. However, wouldn't it be more appropriate for K be scaled such

that flux through a grid cell scales to flux through an idealized channelized system? In other words, I would expect an attempt to scale $\nabla \cdot (T(h) \nabla h)$ with discharge through channels (using an assumed channel spacing). Where the discharge through channels is (e.g. Equation 3 from Hewitt (2013))

$$Q = -K_c S^{(5/4)} \left| \frac{\partial \phi}{\partial s} \right|^{(-\frac{1}{2})} \frac{\partial \phi}{\partial s}$$

**We assume a parameterization for our v_melt term following de Fleurian et al. 2016, where the discharge is given by Darcy's law as $Q = T \nabla h$ (also see our new appendix on the derivation of the parametrization). The channel equations are not applicable in our case since they would describe the effective behaviour of the drainage system in a principally different way from the Darcy law used in our model.**

5) When channelization is introduced into models, they can grow unstably and dominate the system as effective pressures in channels decreases with increasing input. This doesn't appear to occur in your model. You should give a physical description of the terms (and point out the terms in your equations) preventing this.
**This is an excellent point that, in fact, demonstrates one of the advantages of our modeling approach vs. explicitly resolving the channel networks. Just as pointed out in the reviewer's remark, increasing input leads to merging and thus to a lower flow resistance, this positive feedback can easily lead to instabilities. Due to combining unconfined with confined aquifers, an increase in input can be readily redistributed to available empty space in the unconfined parts (or to the outflow boundary if the aquifer is fully confined). Thus $P_w$ rarely rises above $P_i$ in Eq. 8, and this means that $v_{creep}$ provides a good control on runaway hydraulic conductivity.**

6) The argument that the amount of water released from an unconfined aquifer is larger than a confined aquifer is counterintuitive (P4L5). If the head drops 1m in the unconfined case, than wouldn't the water released be much greater than in the confined case, due to the volume of water in the latter being limited by the porosity?
**See our answer to 9) in the first review.**

**7) Numerics**

Your conductivity doesn't appear to show grid convergence, even at resolution of 500m. However, I would expect large scale model runs to require convergence at much coarser resolutions.
**With the updated solutions (see next answer) we have much better grid convergence.**

8) While you identify the conditions under which you get the checkerboard pattern, you don't really explain what is causing it to form. Is this not an artificial pattern due to the numerics? You're solving a highly non-linear equation. In a second order discretization, the dominant truncation error is odd (third order), and hence the error generally will behave in a dispersive manner. Equations 7 and 8 suggest that N ~ Pi_i − h. Since N

appears smooth in your plots, this implies h should be smooth, and then I'm uncertain why K isn't smooth as well.

**We have found the cause of the oscillations: See the response to 11) in the first review. We updated our paper accordingly.**

**9) Application to NEGIS**

Your study domain encompasses areas of both fast flow and slow flow. While the assumptions of the SSA are valid for the ice stream itself, the SSA is not the right approximation for ice flow over the majority of the domain. This is evident in Fig 10a, where we can see that your modelled ice speed is ~1 m/a over the majority of the domain. Over this part of the domain, internal deformation is the key component of ice flow. The comparison of panels A and B in Figure 10 shows not only the effect of subglacial hydrology, but also of different ice physics. Aschwanden et al. (2016) use a 'hybrid' model described in Bueler and Brown (2009), which is itself an approximation (in effect) to the Blatter-Pattyn approximation. It is worth noting that other 'hybrid' approximations exist, such as L1L2 or that of Goldberg (2011). Because of the different regimes in your domain, I think it is necessary to either use a hybrid approximation (if it's available in ISSM), or Blatter-Pattyn to test the results of your coupled model.

**We fully agree with the reviewer, that SSA is not valid for the whole modeling domain. Either a so-called hybrid model (which is not available in ISSM) or a HO model would be more appropriate. However, we think just for the sake of demonstration that our simple approach is fair. In particular, we aim to show an improvement in the fast flow regions where the SSA is valid.**

**The PISM results are shown for completeness and we agree that they are not directly comparable to the ISSM results as the physics etc. differ in too many aspects. In the new version of the manuscript we have now better motivated and clarified our approach.**

Minor Comments:

**P1L2**: ...'drives freshwater into the ocean'... State the explicit impact of this (e.g. undercutting at calving fronts?)
**We updated this sentence and stated the impact.**
**P1L13-16** This paragraph would benefit from references.
**We have added some references(Lliboutry, 1968; Röthlisberger, 1972**
**Gimbert et al. 2016).**
**P2L1:** 'predominant in alpine glaciers and on the margins of the Greenland': This has more nuance, as channels develop seasonally, and are not predominant year round.
**We added "and do usually develop over the summer season when a lot of melt water is available." in the sentence before.**
**P2L7:** I think Hewitt (2013) should be mentioned here, as should Hoffman and Price (2014)

**We have added the requested references.**

**P2L8:** ...'remarkable results for spontaneously evolving channel networks'. The wording/concept could use clarification

**Changed to "While these models demonstrate immense progress for modelling spontaneously evolving channel networks[...]".**

**P2L27-28:** 'While the assumption ... with lower water input'. Citation needed

**P3L15:** hydraulic head needs a definition

**We added "(water pressure in terms of water surface elevation above an~arbitrary datum; piezometric head)".**

**P3L18:** Eq 2: porosity cancels itself out. Can you confirm that the units match up?

**There was a typo in Eq. 2, missing a '+'. It should read** $\rho_w \omega\, g\, (\beta_w + \frac{a}{w})$ **.**

**P3L19:** the definition of alpha in table 1 reads the 'compressibility of water'. The definitions of alpha/beta_w in table 1 needs to be switched I think.

**This is correct. We have corrected the mixup.**

**P3L25:** although h is defined as the hydraulic head, it appears to be used as the saturated height

**Yes, as soon as the aquifer becomes unconfined, the hydraulic head is the same as the saturated height.**

**P3L28:** In Equation 5, it is unclear why Se(h) in the unconfined case depends on b. When the aquifer is unconfined (say with a saturated thickness of 1m), does it matter if the aquifer thickness is 10m or 20m? To gain a better overview of this formulation, I looked at the ' Groundwater flow equation' page on wikipedia ( https://en.wikipedia.org/wiki/Groundwater_flow_equation ). Although I admit that it is not an authoritative source, the formulation there states that Se(h) = S_s*b in the confined case, and S_e(h) = S'(h) is the unconfined case.

**In the unconfined case, the storage is approximated by the specific yield** $S_y$ **, which is much larger than** $S_s b$ **(for the confined case). Therefore, adding a small term (** $S_s b$ **) to a much larger term (** $S_y$ **) does not really make a difference.**

**P4Fig1:** This should be updated/supplemented to show the physical interpretation of the models with channels.

**We have updated the figure and added a colored section representing the efficient system. Together with the improved motivation it should be clear now, how to understand the representation with channels.**

**P5L17:** I would move discretization to an appendix. It's beneficial to any reader looking to reimplement your model, but not necessary in the main text.

**Done.**

**P7L16:** Can you cite the upcoming results as Author(s) (In Prep)?

**We will do so, as soon as the complete list of authors is known.**

**P8 Table 3:** I think it would be beneficial to discuss these values compared to the inferred hydraulic conductivity values of till: 10^-9 - > 10^-4 m/s (Fountain and Walder, 1998).

**Since we do not assume a till bed or any other specific type of drainage (as we hopefully made clearer now, see e.g. our answer to 13) in the first review), we don't think that we need this comparison.**

**P11L4:** 'less' → 'lower'

**Done.**

**P13L18:** basal topography has no influence at all in the unconfined case?

**Sorry, another typo, it should read "the confined-only solution completely depends on boundary conditions (apart from governing dK/dt)."**

**We added a seconds sentence to explain what we mean in more detail:**

**"The possibility of the aquifer to become unconfined captures the expected behaviour much better: At high water levels, water pressure distribution dominates water transport, while at low levels the bed topography becomes relevant."**

**P17L19:** You need to define N_HUY.

**Done.**

**P18L1:** the quotient of X^-1 and Y^-1 could be simplified to the quotient of Y and X.

**Yes, we have simplified that, see also 16) in the first review.**

**P18L8:** It's important to add a citation here to MacAyeal (1989) and/or Morland (1987). In particular, the term SStA is confusing, since this approximation is often known as the SSA in the community. There is a proliferation of 'hybrid' models now, combining SSA and SIA, so a variation like SStA could be misinterpreted as one of those. I looked up the ISSM documentation to be sure that SStA was equivalent to SSA.

**We agree that our term SStA is misleading as it is referenced as SSA in the corresponding ISSM documentation and references. However, if we are correct SSA is Shallow Shelf Approximation where the basal drag is zero. The Shelfy Stream Approximation (SStA) with the basal drag unequal zero is therefore different to the SSA. These terms are often mixed up in the literature. To be consistent with the ISSM documentation we now use the term SSA and give the corresponding citation.**

**P18L31:** The improvement/comparison of your results should be quantified.

**See our answer to 17) in the first review.**

**P19L8.** 'This decisively illustrates the importance of having a real two-way coupling between the ice model and the basal hydrology model in order to obtain good results.' I don't believe you have shown this. You would have to show that you cannot reproduce the results with 2-way coupling, and then show that you reproduce the results when 2-way coupling is introduced. Since it is not cited, I would point the authors to Hoffman and Price (2014) for a detailed discussion on coupling of ice flow and subglacial hydrology.

**In our opinion this constitutes another point in favour of including real two-way coupling. We agree that this is statement was a bit bold and we changed it accordingly in the manuscript.**

**P19L17:** This sentence reads oddly.

**Changed 'restitution' to 'decline'.**

**P24Fig10.** The descriptions and panel labels are mixed up [there is no (d) in the figure caption]

**We have added (d) to the caption.**

[revised manuscript text omitted]

---

## Referee Report (RR1)

**Review of 'A confined-unconfined aquifer model for subglacial hydrology and its application to the North East Greenland Ice Stream', v2**

Doug Brinkerhoff

**Specific Comments**

**Physical Validity**

The authors have addressed many of my original complaints, and I am convinced that their model is implemented and solved correctly. However I am still not convinced that this model is physically justified. The authors give the classic derivations for the equations that govern changes in channel area by dissipative heating and by creep closure. These are not at all controversial. What is controversial is implicitly stating that

$$K = A_c,$$

or that bulk hydraulic conductivity is equal to the area of a hypothetical channel. These things don't even have the same units. $A_c$ is cross sectional area with units m$^2$. $K$ is hydraulic conductivity, with units m s$^{-1}$. They clearly cannot be equal, yet looking at equation A10 versus A11, that is apparently the exact assumption that is being made. It also implies that the first term on the right hand side of Eq. A11 has incorrect units (m s$^{-1}$, while the other two terms have units m s$^{-2}$). Even if a constant were included to deal with the unit problem, I still need to understand why hydraulic conductivity should scale linearly with channel cross-sectional area, when it's typically thought that transmissivity is a non-linear function of channel cross-sectional area.

**Inclusion of sliding**

Fowler (1987) does not necessarily justify the neglect of cavity opening, particularly near the margins. The specific line in Fowler (1987) stating that cavitation is unlikely for ice sheets also assumes low water pressure, which is likely not the case for NEGIS. Similarly, it is mostly unknown whether bedrock undulations are actually longer wavelength in the sub-stream environment than elsewhere. The low slope component of Fowler's conclusion (namely that slopes are around $10^{-3}$ are also not valid here, especially near the margins, where surface slopes are closer to $2.5 \times 10^{-2}$, not so different from a mountain glacier. Finally, there

are many observations from boreholes in Greenland that seem to be consistent with a linked cavity system. Ignoring it seems unjustified to me, regardless of the lack of parameter knowledge.

**Parameter Choices**

I cannot accept the conclusion that because the system being modelled is imaginary, that including parameters from two other physically justifiable models at the same time somehow makes this model capable of capturing the behavior of both.

**Technical corrections**

**P19L14** What does 'empirical nature' mean? Doesn't look like the correction made it into the next version.

---

## Editor Decision (ED1)

Dear Sebastian,

I have now had a chance to look at the revised version of the manuscript. Thank you for including the additional descriptive material on the physical rationale behind the model. This gives a clearer picture of how the model works and I think it does provide an avenue for us to take the manuscript forwards. However, I still do not think we can publish the manuscript in current form. The main reason is that I think you still need to include additional explanatory material on the reasons why you are using the equations that you are using. I have gone into further details of these below.

The important question raised by both referees is why you are using channel opening and closing equations to evolve the conductivity (now transmissivity). In the current version you are prescribing a conductivity and evolving the transmissivity. To do this you are appealing to the de Fleurian et al. (2016) study, where the same equations are used to evolve the thickness of an Equivalent Porous Layer. However, I still think that the model description in the manuscript will cause confusion to many readers, unless you spell out the assumptions behind your approach more fully.

In their study, de Fleurian et al. (2016) do not go into much detail in how they translate evolution equations for channel cross-sectional area into an equivalent porous layer thickness. However, their formulation can be recovered quite straightforwardly under the following assumptions:

1) Channels have area $S_c$ and are spaced apart by $L_c$.
2) EPL thickness is identified as the average water thickness, given by the average cross-sectional area of channels per unit length (along a line perpendicular to channels), which is $d_{EPL} = S_c / L_{c..}$
3) $S_c$ evolves according to the channel opening and closing equations.
4) $L_c$ remains constant.

For Darcy flow with a particular conductivity and pressure gradient, the water flux is proportional to the cross-sectional area, so channels of cross sectional area $S_c$, spaced at distance $L_c$, carry the same flux as the equivalent porous layer of thickness $d_{EPL}$. The assumption of Darcy flow through channels is somewhat unconventional in the glaciological literature, where the Darcy-Weisbach equation is often used. Nevertheless, I think that the explanation above is enough to consider the de Fleurian et al. (2016) interpretation as a physically-motivated model, particularly if channels are assumed to contain sediment, debris or other obstacles, so that Darcy flow is an appropriate assumption.

Now. In the current manuscript, you are evolving the transmissivity ($T$) using the same equations, but you are simply replacing Equivalent Porous Layer thickness with transmissivity. I think this is the source of the difficulty we have all had in understanding the manuscript (in which I include both referees and myself). Unless this is clarified I think most readers will also be confused. You are appealing to the de Fleurian et al. (2016) model (in which $d_{EPL}$ can be conceptualised, as above, as the thickness of a water film that would be produced if all the water in the channels were distributed uniformly). But, it is clear from the description that you are assuming that this film acts as though it is filled with material that has a conductivity $K$, and that this takes the same value as is used for the groundwater flow through the aquifer. In effect then, you are using the channel evolution equations to bring about new aquifer that can transmit flow (channel opening terms), or to remove aquifer (channel closing terms). This is only affecting the transmissivity ($T$). For all other purposes, such as storage, the thickness of the aquifer ($b$) is being kept fixed.

Perhaps the best route forwards is to explain more fully some of the steps that lead to your system of equations. I give two examples of how this could be done below, but I really mean these to illustrate the level of detail needed, not to be prescriptive about the interpretation of the model.

One option would be to include an equation where Transmissivity ($T$) is written as the sum of two terms, as appropriate for flow in parallel through the aquifer ($T_a$) and the channel system ($T_c$).

$$T = T_a + T_c.$$

Unconfined case: $0 < \psi < b$, with $\psi = h - z_b$,

$$T_a = K_a\ \psi,$$

$$T_c = 0,$$

where $K_a$ is the conductivity of the aquifer.

Confined case: $\psi > b$,

$$T_a = K_a\ b,$$

$$T_c = K_c\ d_{EPL,}$$

where $K_c$ is the conductivity of the equivalent porous layer (or equivalently of the Darcy flow through channels of cross-sectional area $S_c$, spaced apart by distance $L_c$).

As above, the Equivalent Porous Layer ($d_{EPL}$) for fixed channel spacing ($L_c$) is

$$d_{EPL} = S_c / L_c.$$

Channel cross-sectional area ($S_c$) evolves (as already described in the appendix),

$$d(S_c)/dt = Melt + Cavity\_opening - Creep\_closure$$

The system that you are solving appears to assume that conductivities are the same in the aquifer and in the equivalent porous layer, so that $K_c = K_a$. You also seem to be neglecting flow in the aquifer for the confined case, so that $T = T_c$, in the confined case, rather than $T = T_a + T_c$.

If you agree with this interpretation of your equations then I think you need to include these arguments, and these additional steps, to the model derivation in the manuscript (using your own preferred notation for the quantities referred to above), otherwise the manuscript presents no logic as to why you are solving the system that you are solving. If you do this, then the assumptions need to be justified. The assumption $K_c = K_a$ is perhaps appropriate if channels are filled with sediment, but you will then need to explain that any effect of the sediment on creep closure has been neglected. The assumption $T = T_c$ in the confined case could perhaps be justified if $d_{EPL} >> b$, and you could test this from analysis of your existing results.

One problem with the above interpretation is that your model only seems to include storage in the aquifer, not in the equivalent porous layer, but if $d_{EPL} >> b$ this does not seem appropriate. An alternative interpretation, is that the equivalent porous layer is thinner than the aquifer, so that $d_{EPL} << b$, allowing you to neglect storage in channels. If channels are much more conductive than the aquifer, so that $K_c >> K_a$, then it is possible that $T_c >> T_a$, so that $T$ is approximately proportional to $d_{EPL}$ in the confined case, despite the equivalent porous layer being thinner than the aquifer. This solves the storage problem, but does not explain why you are using one value of $K$ throughout. In that case, you should be using $K_c$ in the melting term, and the cavity opening term, so, under this interpretation, you are either underestimating these terms, or overestimating flux through the

aquifer. The cavity opening term could be dealt with simply by changing the cavitation step height, so that the parameter $\beta$ is unchanged from your simulations, but that still leaves either the melt term underestimated, or the flux through the aquifer overestimated.

If you do not agree with either of the above interpretations, then you will need to supply a similarly detailed picture of how you consider that the model can be derived from some physical picture of the system under consideration. The description should make it clear what assumptions have been made and what the consequences of those assumptions might be. Unless I feel that this description has been provided I will reject the manuscript. Simply appealing to similarities with de Fleurian (2016) study, as you do in the present version of the manuscript, is not enough. It does not provide enough guidance for the readers to assess whether the model can be expected to behave realistically or not.

To be more specific, the main difficulties are with the derivation in the appendix.

P18. Equation A2. As described above, you need to provide a physical interpretation here that makes sense. This interpretation says that aquifer thickness grows when channels grow in area, but melting does not bring a new layer of aquifer into existence. I think you need to separate out aquifer thickness $b$ and Equivalent Porous Layer thickness $d_{EPL}$ and be much clearer about which concept is in use at each stage. Please think carefully about this and present a coherent explanation for the model equations. You cannot replace one quantity (channel area $A_c$) with another (aquifer thickness $b$) unless you provide some physical reasoning why you are doing this.

P19. Equation A7. Same problem. Please provide an explanation that has some physical reasoning behind it. You need to be clearer about what is aquifer thickness ($b$) and what is equivalent porous layer thickness ($d_{EPL}$). It is the latter that is controlled by the opening equations. Creep closure of channels is not usually considered to destroy aquifer.

P19. Line 17. Same problem. Please provide an explanation that has some physical reasoning behind it.

P20. Line 5. Same problem. Please provide an explanation that has some physical reasoning behind it. This only applies if aquifer thickness $b$ is changing, but I think it is the thickness of the Equivalent Porous Layer that is changing.

Please include the cavity opening term and give the reason why it takes the form that it does. Cavity opening creates channel area at rate $v_b\, h_{step}$, where is $v_b$ is sliding speed and $h_{step}$ is step height. This provides a source of channel area, not a source of aquifer thickness. However, if $T_c = K_c\, d_{EPL}$ and $d_{EPL} = S_c / L_c$ a cavity opening term of similar form can be recovered. Please go through the steps needed to relate cavity opening to channel area and transmissivity and include this chain of reasoning in the manuscript.

To be clear. I will reject the paper if these questions are not clarified. I don't think this needs to happen, because I think there are conditions (as outlined above) for which the system of equations that you are solving, or perhaps a slight modification of them using two conductivities ($K_c$ and $K_a$), can be justified. You need to do a much better job at explaining the physical motivation behind the model in the manuscript.

I have also included some more minor technical corrections below.

Yours sincerely,

Robert Arthern

Minor technical corrections

Please go through carefully and check which equations should be using the hydraulic head (h) and which should be using the relative value (psi).

In particular,

   i)      Equation 4. Shouldn't $\psi$ be used to determine whether the system is confined or not.
   ii)     Equation 7. Shouldn't pressure be $P = \rho_w\, g\, h$, not $\rho_w\, g\, \psi$.

Appendix A.

P19. Please correct description of chain rule. There is a missing value of $R_c$.

P19. Probably better to leave the sign on the gradient ($G$) and the flux ($q$) rather than taking magnitudes. If you do this, please go through carefully and make sure melt term is defined to have the correct sign.

P19. flux per unit length(?).

---

## Author Response (AR2)

Dear Editor, dear Reviewers,
The authors wish to thank you again for valuable remarks and patience. We fully understand your request to provide a clear physical justification for our modeling approach. In order to address this issue we adopted a more general (transmissivity) formulation for our aquifer model that can be directly derived (by multiplication with the constant hydraulic conductivity K) from the corresponding evolution equation for aquifer thickness of De Fleurian et al. 2016 or by multiplying with constant aquifer thickness b from our formulation for the hydraulic conductivity. Using this generalized model, we adopted the same set of assumptions (K=const) as in De Fleurian et al. 2016 instead of (b=const) used in our previous revision (we also thank the reviewers for pointing out a problem with units). By doing this, we ensure that we use the exact same parameterization as the original works -- as we intended -- and simplify the comparison of the results. The generalized model retains all the features (single layer for effective/ineffective system, combination of confined and unconfined Darcy flows) of our model presented in earlier revisions. We have re-run all our experiments, because we had to adjust the parameters, but the basic model is still the same as before. This new round of tuning was also necessary, because we added -- as Referee #1 suggested -- a term representing cavity opening into the model.

**Referee #1:  Doug Brinkerhoff**

**Physical Validity**

The authors have addressed many of my original complaints, and I am convinced that their model is implemented and solved correctly. However I am still not convinced that this model is physically justified. The authors give the classic derivations for the equations that govern changes in channel area by dissipative heating and by creep closure. These are not at all controversial. What is controversial is implicitly stating that $K = A_c$, or that bulk hydraulic conductivity is equal to the area of a hypothetical channel. These things don't even have the same units. $A_c$ is cross sectional area with units m². K is hydraulic conductivity, with units m s⁻¹. They clearly cannot be equal, yet looking at equation A10 versus A11, that is apparently the exact assumption that is being made. It also implies that the first term on the right hand side of Eq. A11 has incorrect units (m s−1, while the other two terms have units m s−2). Even if a constant were included to deal with the unit problem, I still need to understand why hydraulic conductivity should scale linearly with channel cross-sectional area, when it's typically thought that transmissivity is a non-linear function of channel cross-sectional area.

After carefully checking our equation again, we found that we have made an error in the conversion to conductivity. We fixed this error and decided to generalize the formulation by evolving the transmissivity instead of the hydraulic conductivity. This re-formulation allows us to confirm the reviewer's claim of non-linear dependence between the transmissivity and the channel cross-section that is somewhat awkward to handle numerically. We came to the

conclusion that it is simpler if we use the original equation of de Fleurian et al. 2016 and evolve the transmissivity under assumption (K=const) resulting in a linear dependence.

**Inclusion of sliding**

Fowler (1987) does not necessarily justify the neglect of cavity opening, particularly near the margins. The specific line in Fowler (1987) stating that cavitation is unlikely for ice sheets also assumes low water pressure, which is likely not the case for NEGIS. Similarly, it is mostly unknown whether bedrock undulations are actually longer wavelength in the sub-stream environment than elsewhere. The low slope component of Fowler's conclusion (namely that slopes are around $10^{-3}$ are also not valid here, especially near the margins, where surface slopes are closer to $2.5 \times 10^{-2}$, not so different from a mountain glacier. Finally, there are many observations from boreholes in Greenland that seem to be consistent with a linked cavity system. Ignoring it seems unjustified to me, regardless of the lack of parameter knowledge.

We realize that it really is a good idea to include sliding/a cavity opening term and did so in our evolution equation. The advantage is, that now we do not need to use the minimum for the conductivity/transmissivity to have an initial opening and preventing the system from going completely watertight, but have a proper term to control this.

**Parameter Choices**

I cannot accept the conclusion that because the system being modelled is imaginary, that including parameters from two other physically justifiable models at the same time somehow makes this model capable of capturing the behavior of both.

Whereas there is a clear physical justification of the ineffective model, the representation of channels using the Darcy model relies on a wide array of assumptions (especially those concerning the channel geometry and size distribution) that are not so easy to consider realistic. Our idea of creating an effective model is to use a somewhat modified set of assumptions to represent large scale (bulk) behavior for both types of drainage. Similar approaches are widely used for applications in subsurface modeling (see, e.g., double porosity models for fractured aquifers). Certainly the way the porosity is treated is different in a channelized system than in an inefficient system, but the proposed approach assumes that an active effective system clearly dominates the flow thus the choice of parameters in our model is clearly biased towards the effective type of flow. For ineffective flow situations dominated by 'background' conductivity, the model uses limiting values T_min with the corresponding classical Darcy behavior -- just as the separate ineffective layer models do in two-layer approaches.

**Technical corrections**

**P19L14** What does 'empirical nature' mean? Doesn't look like the correction made it into the next version.

We wanted to express our idea to use the available data to build the simplest possible model and we have updated the sentence like this:

> *Since extensive observations of the subglacial system are rare, its large scale approach to fit a simple parametrization of the flow to the available data can be an advantage.*

**Referee #2**

**1.** In the opening comments, the authors better describe the aim of the model:

> *"Our goal is not to fit an aquifer model to the really existing subglacial aquifers by finding correct geometrical and physical parameters. This approach would likely work for flows through till or similar types of drainage but just as likely fail for channel networks. Instead we chose Darcy model as an 'idealized' representation of all drainage processes (including channels) and tuned it to give the best fit to the available data."*

A few sentences down, they remark that to better explain this point, they have added a corresponding sentence in the introduction. However, the sentence I think they're referring to isn't sufficient. I think it would be best for the authors to add a new paragraph, essentially laying out clearly their in motivation as in the above quote. This would really avoid confusion for readers.

This is a good point and we modified the introduction to attempt to better explain our motivation and modeling assumptions.

**2.** I'm not certain I fully understand the author's reply to my point 5 (regarding instability with channels). Further the authors have chosen to add a sentence P19L15 in the conclusion stating that no instability is a positive aspect of their model. I think this requires them to add a corresponding section in the discussion about the instability, since the conclusions should cover the discussion (unless I missed it there). This would be a good opportunity to try to be more lucid about what's going on. Can this be related to physics, or is this an outcome of the model parameterization?

The reason for why we do not have this instability is, that in the parametrization that we adopted from De Fleurian et al. 2016 the melt term, as well as the creep term depend linearly on the transmissivity/layer thickness. This means, that there is no potential for instability, because the melt term does not grow way faster than the creep term, when it is slightly perturbed. The instability usually occurs when a turbulent flow parametrization, such as Darcy-Weisbach is used (see e.g. Schoof 2010).

We added a small paragraph explaining this in the paper.

**3.** For my point number 6, the authors direct me to the response of point number 9 of the other reviewer. I'm not clear what their response exactly means, except that perhaps this goes back to the point that there model isn't meant to be physical. They perhaps need to do a better job guiding us to what parts of the model are seen as physical, and what parts are parameterizations tuned for the model to fit other models/reality.

> ***First review point 6***: *The argument that the amount of water released from an unconfined aquifer is larger than a confined aquifer is counterintuitive (P4L5). if the head drops 1m in the unconfined case, than wouldn't the water released be much greater than in the confined case, due to the volume of water in the latter being limited by the porosity?*

We think we have a (likely somewhat syntactic rather than in substance) misunderstanding regarding the role of the storage coefficient in both types of model. In the unconfined case, the water can be stored in the empty pore space with little effort (i.e. small changes in head result in large changes in the volume of stored water). In the confined case, all pore space is filled, some additional water can be stored by compressing the solid matrix usually possible due to trapped air; this type of storage increase requires a lot of effort (i.e. small changes in head cannot produce large changes in volume of stored water). This is the meaning behind $S_s b \ll S_y$ expression in our previous response. Also see Bear, Cheng (Springer, 2010, p. 214) for more detail.

**4.** I'm not certain that keeping the SSA is in the best interest for the manuscript. It seems like an poor choice, especially since there doesn't seem to be a computational limitation for using a higher order model. I wouldn't derail the paper over this, but I wouldn't think that rerunning the model would be that hard. This effects the author's misfit quantification, because the SSA model clearly fails in the slow moving regions, while improves when hydrology is introduced. If they use Blatter-Pattyn, the misfit would be lower in the no-hydrology case for slow moving regions, and then the fit may decrease over this region when hydrology is introduced. Overall however, since the fit leaves room for improvement, I suspect most suspect readers will only go as far as interpreting the results visually.

We agree that this is a valid point and we have repeated our experiments using higher order equations (Blatter-Pattyn). Due to the additional deformation part, the overall velocity increases (more visible in the no-hydrology case, because of the color scale) and the modelled velocity is closer to the observed data. The result is still the same: using hydrology improves the RMS by about 50 m/a. We updated the text in the manuscript to reflect these changes.

[revised manuscript text omitted]

---

## Author Response (AR3)

Answer to Editor

Dear Editor,
Please accept our sincere gratitude for taking your time to provide a detailed analysis of the unclarities and inconsistencies in the previous version of our manuscript. It is obvious that our manuscript lacked sufficient details on the justification of the physical and mathematical model. With the new version, we hope to have addressed these issues and took pains to explain our motivation and approach as well as include new parts to provide more information on our assumptions and the derivation of the equations.
We think that there was a general misconception with regard to the total number of layers in our model (we only have a single one representing both flow systems). Analysing the sources for this misunderstanding, we think we found and resolved those issues: After the introduction, we presented our model equations accompanies by a reference to deFleurian(2014/16) with little additional explanation. Even worse, our derivation/explanation of the evolution equations in the appendix was incomplete, and we were not quite consistent in our notation ($b$ and $d_{EPL}$). We

reorganized and extended our methods section to make our assumptions clear. We further improved our appendix with the detailed derivation of the equations, while explaining how the change in channel area (and cavity-space) translates to changes in transmissivity.

To summarize the most important points:
- We model the whole system of efficient and inefficient drainage with a single aquifer layer (equivalent porous medium (EPM)) by locally adjusting its transmissivity.
- Although usually described by the Darcy-Weisbach equation, we approximate fast flow through the efficient system by Darcy flow with high effective transmissivity.
- We derive the temporal evolution of the controlling parameter ---effective transmissivity--- from the temporal evolution of the volume occupied by channels (deFleurian 2016) and cavities (Werder 2013).
- The unconfined formulation (Ehlig and Halepaska, 1976) is a necessary addition to obtain physical water pressures, and, therefore, we moved it to the end of the methods section. We do not consider the evolving transmissivity in case of unconfined flow and also ignore unconfined flow when computing the melt rate (This is now described at the end of section 2.2).

We also changed the sketch showing the modelling concept to make it clear that we have only a single layer of an equivalent porous medium.
Below we answer each of your points and mark our answer in black, whereas the original comment is shown in grey.

I have now had a chance to look at the revised version of the manuscript. Thank you for including the additional descriptive material on the physical rationale behind the model. This gives a clearer picture of how the model works and I think it does provide an avenue for us to take the manuscript forwards. However, I still do not think we can publish the manuscript in current form. The main reason is that I think you still need to include additional explanatory material on the reasons why you are using the equations that you are using. I have gone into further details of these below.

Thank you for your comments. We worked through the manuscript to eliminate the weak points in the argumentations. Again, we think the confusion is mainly caused by the comparison with methods in DeFleurian et al. (2014, 2016, the double continuum model). Although both models share a number of assumptions, they differ in two main aspects:

- We only have one "layer" (in total) of an equivalent porous media (EPM) responsible for efficient and inefficient water transport.
- We consider in addition the unconfined situations to avoid negative water pressure.

Another source of confusion was that we mistakenly used $b$ for two different quantities, one in the Appendix and one in the main text - our apologies!

The important question raised by both referees is why you are using channel opening and closing equations to evolve the conductivity (now transmissivity). In the current version you are prescribing a conductivity and evolving the transmissivity. To do this you are appealing to the de Fleurian et al. (2016) study, where the same equations are used to evolve the thickness of an Equivalent Porous Layer. However, I still think that the model description in the manuscript will cause confusion to many readers, unless you spell out the assumptions behind your approach more fully.

We have rephrased and restructured our methods section and the Appendix accordingly (see also our introductory reply above).

In their study, de Fleurian et al. (2016) do not go into much detail in how they translate evolution equations for channel cross-sectional area into an equivalent porous layer thickness. However, their formulation can be recovered quite straightforwardly under the following assumptions:

1)  Channels have area $S_c$ and are spaced apart by $L_c$.

Channel spacing is not considered in the model. The contribution of the channel network to the large-scale/average change in equivalent transmissivity can be caused by the increase/decrease of cross-sectional area of one, some, or many channels or just by the variation in the number of channels in the gridbox. This is not explicitly accounted for in the equivalent porous media (EPM) approach.

2) EPL thickness is identified as the average water thickness, given by the average cross-sectional area of channels per unit length (along a line perpendicular to channels), which is $d_{EPL} = S_c/L_c$.

The equivalent porous media thickness $b$ is a model parameter that has been adjusted

according to the SHMIP/B2 simulations (see details given in section 3.1). We use $b$ (subscript

EPM is omitted) instead of $d_{EPL}$ according to the original nomenclature used in Ehlig &

Halepaska (1976) for the confined/unconfined formulation. Evolving the transmissivity via the volume occupied by the channel network can be translated into an average water thickness (channels only) per unit area. This is outlined in Appendix A and is closely related to the evolution of the EPL thickness given in DeFleurian et al. (2016, Eq. 6). Additionally, we consider cavity opening.

3) $S_c$ evolves according to the channel opening and closing equations.

4) $L_c$ remains constant.

We extended our appendix in order to improve clarity. Also see our answers to your point 2) above.

For Darcy flow with a particular conductivity and pressure gradient, the water flux is proportional to the cross-sectional area, so channels of cross sectional area $S_c$, spaced at distance $L_c$, carry the same flux as the equivalent porous layer of thickness $d_{EPL}$. The assumption of Darcy flow through channels is somewhat unconventional in the glaciological literature, where the Darcy-Weisbach equation is often used. Nevertheless, I think that the explanation above is enough to consider the de Fleurian et al. (2016) interpretation as a physically-motivated model, particularly if channels are assumed to contain sediment, debris or other obstacles, so that Darcy flow is an appropriate assumption.

The idea of an equivalent porous medium model is that, by adjusting the properties, one can mimic the effective behaviour of the more complex medium. The model does not represent water flow through individual channels (which would be better represented by Darcy-Weisbach) and, therefore, Darcy is the appropriate constitutive law. It is not assumed that channels contain sediment, debris, or other obstacles (but they certainly may contain any of that).

Now. In the current manuscript, you are evolving the transmissivity ($T$) using the same equations, but you are simply replacing Equivalent Porous Layer thickness with transmissivity. I think this is the source of the difficulty we have all had in understanding the manuscript (in which I include both referees and myself). Unless this is clarified I think most readers will also be confused. You are appealing to the de Fleurian et al. (2016) model (in which $d_{EPL}$ can be conceptualised, as above, as the thickness of a water film that would be produced if all the water in the channels were distributed uniformly). But, it is clear from the description that you

are assuming that this film acts as though it is filled with material that has a conductivity $K$, and that this takes the same value as is used for the groundwater flow through the aquifer. In effect then, you are using the channel evolution equations to bring about new aquifer that can transmit flow (channel opening terms), or to remove aquifer (channel closing terms). This is only affecting the transmissivity ($T$). For all other purposes, such as storage, the thickness of the aquifer ($b$) is being kept fixed.

On the broad scale you are right: we use channel evolution equations (and cavity opening term) to compute the change of available volume in the drainage system, which is translated into an effective transmissivity. For the resulting flux or head it makes no difference if the additional volume is introduced in terms of layer thickness (addition or removal aquifer) or if we translate it into transmissivity (which could be seen as an increase in relative conduit space inside the layer of the equivalent porous medium).

We do not discriminate between an inefficient and an efficient systems, and model just a single system that evolves according to all respective processes (melt, cavity opening, creep). We would argue that this is a more natural description than the usual separation between the two systems. We assume that the difference between the two systems can be expressed by a locally adjusted variable ($T$ in our case). Our results show that, on a large scale, this is reasonable, and we can approximate the important behaviour (effective pressure) of subglacial flow.

The addition of the unconfined flow is necessary to obtain physically meaningful values for water pressure in some situations (namely, low water supply, which can violate the assumption of the water system being always filled).

Perhaps the best route forwards is to explain more fully some of the steps that lead to your system of equations. I give two examples of how this could be done below, but I really mean these to illustrate the level of detail needed, not to be prescriptive about the interpretation of the model.

One option would be to include an equation where Transmissivity ($T$) is written as the sum of two terms, as appropriate for flow in parallel through the aquifer ($T_a$) and the channel system ($T_c$).

$$T = T_a + T_c.$$

Unconfined case: $0 < \Psi < b$, with $\Psi = h - z_b$,

$$T_a = K_a \Psi,$$
$$T_c = 0,$$

Where $K_a$ is the conductivity of the aquifer.
Confined case: $\Psi > b$,

$$T_a = K_a b,$$
$$T_c = K_c d_{EPL},$$

where $K_c$ is the conductivity of the equivalent porous layer (or equivalently of the Darcy flow through channels of cross-sectional area $S_c$, spaced apart by distance $L_c$).

As above, the Equivalent Porous Layer ($d_{EPL}$) for fixed channel spacing ($L_c$) is
$$d_{EPL} = S_c/L_c .$$
Channel cross-sectional area ($S_c$) evolves (as already described in the appendix),
$$dS_c/dt = Melt + Cavity\_opening - Creep\_closure$$

The system that you are solving appears to assume that conductivities are the same in the aquifer and in the equivalent porous layer, so that $K_c = K_a$. You also seem to be neglecting flow in the aquifer for the confined case, so that $T = T_c$, in the confined case, rather than $T = T_a + T_c$.

In our implementation of the equivalent porous media approach, we indeed assume that the equivalent transmissivity can be composed from two contributions: the transmissivity of the background material, $T_a$, and the equivalent transmissivity of the conduits (channels and cavities), $T_c$ and thus, $T = T_a + T_c$. This is independent from the confined/unconfined question. We may have not made it clear enough, but we assume, that changes in the equivalent transmissivity over time are driven by both, the channel system and cavities. Thus
$$\frac{dT}{dt} = \frac{dT_a}{dt} + \frac{dT_c}{dt}$$
The time independent (supply and ice sheet basal sliding independent) contribution $T_a$ from the background material is very similar to our Tmin value that is used as a model parameter. We show that the model is not very sensitive on the choice of Tmin.
An increase of the cross section of one channel (or several smaller channels) or an increase of the number of channels (decrease of channel spacing, $L_c$) within one grid cell translates into an increase of $T_c$ and thus $T$. We therefore apply the channel evolution equation detailed in the appendix for the evolution of $T_c$ (similar for the cavities). In short: $T = T_a + T_c$, but $dT/dt \approx d(T_c)/dt$ in the model. Thus, $dT / dt$ is driven by the supply and sliding dependent contributions in the transmissivity. The channel evolution and thus the changes in the effective part of the hydrological system contribute the most to the changes in effective transmissivity of the EPM.

If you agree with this interpretation of your equations then I think you need to include these arguments, and these additional steps, to the model derivation in the manuscript (using your own preferred notation for the quantities referred to above), otherwise the manuscript presents no logic as to why you are solving the system that you are solving. If you do this, then the assumptions need to be justified.
The text has been changed according to the arguments above.

The assumption $K_c = K_a$ is perhaps appropriate if channels are filled with sediment, but you will then need to explain that any effect of the sediment on creep closure has been neglected. The assumption $T = T_c$ in the confined case could perhaps be justified if $d_{EPL} \gg b$, and you could test this from analysis of your existing results.
We do not assume $K_c = K_a$ or $T = T_c$. See comments above.

One problem with the above interpretation is that your model only seems to include storage in the aquifer, not in the equivalent porous layer, but if $d_{EPL} \gg b$ this does not seem appropriate. An alternative interpretation, is that the equivalent porous layer is thinner than the aquifer, so that $d_{EPL} \ll b$, allowing you to neglect storage in channels. If channels are much more conductive than the aquifer, so that $K_c \gg K_a$, then it is possible that $T_c \gg T_a$, so that $T$ is approximately proportional to $d_{EPL}$ in the confined case, despite the equivalent porous layer being thinner than the aquifer. This solves the storage problem, but does not explain why you are using one value of $K$ throughout. In that case, you should be using $K_c$ in the melting term, and the cavity opening term, so, under this interpretation, you are either underestimating these terms, or overestimating flux through the aquifer. The cavity opening term could be dealt with simply by changing the cavitation step height, so that the parameter is unchanged from your simulations, but that still leaves either the melt term underestimated, or the flux through the aquifer overestimated.

As described before, we only have a *single* system/layer representing the complete drainage system, which also means that there is only a single storage mechanism. In our model, the storage ($S = S_s b$) depends on a material property ($S_s$) and the layer thickness ($b$), which is constant. For the unconfined case, the storage is greater resulting in slower pressure changes. Other subglacial hydrology models (e.g. Werder 2013) typically use a storage term that depends on the water pressure, but it is generally a poorly known quantity. We have some hope that in the future remote sensing and ground penetrating radar may shed some light into water storage, but this may still have a long way to go.

If you do not agree with either of the above interpretations, then you will need to supply a similarly detailed picture of how you consider that the model can be derived from some physical picture of the system under consideration. The description should make it clear what assumptions have been made and what the consequences of those assumptions might be. Unless I feel that this description has been provided I will reject the manuscript. Simply appealing to similarities with de Fleurian (2016) study, as you do in the present version of the manuscript, is not enough. It does not provide enough guidance for the readers to assess whether the model can be expected to behave realistically or not.

To be more specific, the main difficulties are with the derivation in the appendix.

P18. Equation A2. As described above, you need to provide a physical interpretation here that makes sense. This interpretation says that aquifer thickness grows when channels grow in area, but melting does not bring a new layer of aquifer into existence. I think you need to separate out aquifer thickness *b* and Equivalent Porous Layer thickness *dEPL* and be much clearer about which concept is in use at each stage. Please think carefully about this and present a coherent explanation for the model equations. You cannot replace one quantity (channel area *Ac*) with another (aquifer thickness *b*) unless you provide some physical reasoning why you are doing this.

We have re-written the appendix and the related parts in the main text to explain the key differences between our EPM layer thickness $b$ and the volume of channels per unit area $b_c$.

The latter is used in DeFleurian et al. (2014, 2016) to evolve the EPL layer thickness ($d_{EPL}$) representing the efficient system only. We assume, that effective transmissivity of our *single* EPM layer increases if the channel volume ($b_c$) increases.

Opening and closure of channels is usually formulated in a 2D cross-sectional point of view. Thus all quantities are expressed in "per unit length", e.g. mass change (Eq. A1). But we need all quantities "per unit area". To give an example: $A_c$ is the channel volume per unit length, thus an area. The same volume per unit area is $b_c$ and thus a thickness (Eq. A2).

You are entirely right here, and we apologize that we overlooked having used $b$ for two different

quantities in the manuscript. With the new version of the appendix this should be clear now.

It is the latter that is controlled by the opening equations. Creep closure of channels is not usually considered to destroy aquifer.
Indeed, creep closure of channels will not destroy the aquifer. Creep closure of conduits (channels or cavities) is considered to reduce the effective transmissivity of the EPM.

See comment to P18. Equation A2.

See comment to P18. Equation A2.

Please include the cavity opening term and give the reason why it takes the form that it does. Cavity opening creates channel area at rate $v_b h_{step}$, where is $v_b$ is sliding speed and $h_{step}$ is step height. This provides a source of channel area, not a source of aquifer thickness. However, if $T_c = K_c d_{EPL}$ and $d_{EPL} = S_c/L_c$ a cavity opening term of similar form can be recovered. Please go through the steps needed to relate cavity opening to channel area and transmissivity and include this chain of reasoning in the manuscript.
The cavity opening term is based on Werder 2013 and reads $\dot{m}_{cavity} = \rho_i \beta |v_b|$, where $\beta = b_r/l_r$ with the typical bump height $b_r$ and distance $l_r$. In this form, the opening take the form of a

change in thickness, which can be translated into volume and then to transmissivity. We have added the description to the manuscript (Section 2.1 and appendix A).

To be clear. I will reject the paper if these questions are not clarified. I don't think this needs to happen, because I think there are conditions (as outlined above) for which the system of equations that you are solving, or perhaps a slight modification of them using two conductivities (*Kc* and *Ka*), can be justified. You need to do a much better job at explaining the physical motivation behind the model in the manuscript.
We fully understand your criticism after going once again through the entire text looking for the sources of confusion about two layers vs. one layer. Likely, after working for a long time on the single layer approach, we've became somewhat desensitized to that. We invested a lot of time to improve this aspekt and all of the authors had their 'oh yes, I can understand what he means' moment.

Minor technical corrections

Please go through carefully and check which equations should be using the hydraulic head (h) and which should be using the relative value (psi).
In particular,

i) Equation 4. Shouldn't $\Psi$ be used to determine whether the system is confined or not.
This is correct and is how it is done in the code. We have fixed the mistake in the manuscript.

ii) Equation 7. Shouldn't pressure be $P = \rho_w g h$, not $\rho_w g \Psi$.
We are certain that the water pressure is $P_w = \rho_w g \Psi$. The water pressure does only depend on the height of the water level above the base and not on the total height:

[Figure]

Appendix A.

P19. Please correct description of chain rule. There is a missing value of $R_c$.
Fixed.

P19. Probably better to leave the sign on the gradient (*G*) and the flux (*q*) rather than taking magnitudes. If you do this, please go through carefully and make sure melt term is defined to have the correct sign.
As the melt depends only on the magnitude we prefer to leave the magnitudes in the text to be keep the nomenclature as in Cuffey and Paterson (2010, Eq. 6.16). Although the signs in G and q would cancel each other out.

P19. flux per unit length(?).
We removed the question mark as it was only a comment for co-authors.

[revised manuscript text omitted]